# A striatal circuit balances learned fear in the presence and absence of sensory cues

Michael Kintscher, Olexiy Kochubey, Ralf Schneggenburger*

Laboratory for Synaptic Mechanisms, Brain Mind Institute, School of Life Science, Ecole Polytechnique Fédérale de Lausanne, Lausanne, Switzerland

**Abstract** During fear learning, defensive behaviors like freezing need to be finely balanced in the presence or absence of threat-predicting cues (conditioned stimulus, CS). Nevertheless, the circuits underlying such balancing are largely unknown. Here, we investigate the role of the ventral tail striatum (vTS) in auditory-cued fear learning of male mice. In vivo $Ca^{2+}$ imaging showed that sizable sub-populations of direct (D1R+) and indirect pathway neurons (Adora+) in the vTS responded to footshocks, and to the initiation of movements after freezing; moreover, a sub-population of D1R+ neurons increased its responsiveness to an auditory CS during fear learning. In-vivo optogenetic silencing shows that footshock-driven activity of D1R+ neurons contributes to fear memory formation, whereas Adora+ neurons modulate freezing in the absence of a learned CS. Circuit tracing identified the posterior insular cortex (pInsCx) as an important cortical input to the vTS, and recording of optogenetically evoked EPSCs revealed long-term plasticity with opposite outcomes at the pInsCx synapses onto D1R+ - and Adora+ neurons. Thus, direct- and indirect pathways neurons of the vTS show differential signs of plasticity after fear learning, and balance defensive behaviors in the presence and absence of learned sensory cues.

*For correspondence:
ralf.schneggenburger@epfl.ch

Competing interest: The authors declare that no competing interests exist.

## Editor's evaluation

This important study examines the contribution of an understudied brain region to fear conditioning in mice. The evidence supporting the authors' conclusions is convincing. This paper will interest neuroscientists working in the fields of basal ganglia, amygdala, and fear learning.

## Introduction

Fear learning is an evolutionary conserved behavior, critically important for animals to detect signs of danger in an ever-changing environment. As such, fear learning is necessary for survival (*Phelps and LeDoux, 2005*; *Janak and Tye, 2015*). Nevertheless, learned defensive behaviors need to be finely regulated, so that animals can return to their normal behaviors after the cessation of threat -predicting sensory cues (*Zanette et al., 2011*). Furthermore, a pathological overexpression of defensive behaviors is a hallmark of several anxiety-related disorders in humans (*Dunsmoor et al., 2011*). Therefore, it is important to understand the neuronal circuits that balance the expression of learned defensive behaviors during and after the presence of threat-predicting sensory cues.

The study of the neuronal mechanisms of fear learning has been strongly facilitated by employing auditory-cued fear learning in model animals like rodents (*Davis, 1992*; *LeDoux, 2000*). In fear learning, subjects learn to associate an initially neutral sensory cue, the conditioned stimulus (CS; often an auditory stimulus), with an aversive or painful outcome like a footshock (the unconditioned stimulus, US). After associative learning, subjects will develop a defensive behavior when the CS is

later presented alone (*Fanselow, 2018*). The defensive behavior that is mostly studied in the context of fear learning in rodents is behavioral arrest, also called freezing (*Fanselow, 1994*). Studies spanning several decades have firmly established that the amygdala has an important role in fear learning (see *Davis, 1992*; *LeDoux, 2000*; *Duvarci and Pare, 2014*; *Tovote et al., 2015*, for reviews). The lateral amygdala (LA) is viewed as an input structure to the amygdalar complex (*LeDoux et al., 1990*), which connects to both the basal amygdala (BA) and the central amygdala, where further integration and processing takes place (*Romanski et al., 1993*; *Quirk et al., 1995*; *Amano et al., 2011*; *Grewe et al., 2017*). Finally, the execution of learned freezing depends on a central amygdala to midbrain (periaqueductal gray) projection (*LeDoux et al., 1988*; *Tovote et al., 2016*). Nevertheless, it is likely that further neuronal circuits beyond these amygdalar circuits are involved in fear learning.

The striatum is part of the basal ganglia motor system, a neuronal system with important roles in the control of movement, action selection, and reward-based learning (for reviews, see *Hikosaka et al., 2000*; *Graybiel, 2005*; *Grillner et al., 2005*; *Redgrave et al., 2010*; *Nelson and Kreitzer, 2014*; *Klaus et al., 2019*). The principal neurons of the striatum are inhibitory projection neurons of two types. First, striato-nigral neurons project in a direct pathway to basal ganglia output structures like the Substantia nigra pars reticularis and others; these neurons selectively express D1-dopamine receptors (D1R). Second, striato-pallidal neurons project in a more indirect pathway towards the basal ganglia output structures; these neurons selectively express D2-dopamine receptors, and also adenosine-2A receptors (Adora) (*Gerfen et al., 1990*; *Schiffmann and Vanderhaeghen, 1993*). Different sub-areas of the striatum have different roles in motor control and motor learning. The dorsal striatum is involved in the learning of motor sequences and in the selection of appropriate actions (*Nelson and Kreitzer, 2014*; *Klaus et al., 2019*), as well as in habit formation (*Redgrave et al., 2010*; *Burguière et al., 2015*), whereas the ventral striatum is important for reward-based learning (*Humphries and Prescott, 2010*; *Cox and Witten, 2019*).

Recently, based on brain-wide studies of the cortical inputs to the mouse striatum, a further sub-area of the striatum was identified; a posterior area called tail striatum (*Hintiryan et al., 2016*; *Hunnicutt et al., 2016*; see *Valjent and Gangarossa, 2021* for review). Interestingly, early in vivo recordings in the LA have found CS- and US-responsive neurons in the tail striatum adjacent to the LA (*Romanski et al., 1993*). Recent work has shown that the tail striatum is, similarly as other striatal areas, composed of D1R-expressing (D1R+) and D2R- and Adora-expressing neurons (*Gangarossa et al., 2019*). Furthermore, in vivo imaging studies have shown that dopaminergic axons in the tail striatum code for salient sensory stimuli, but not for rewarding stimuli (*Menegas et al., 2018*). However, the role of the tail striatum in fear learning has not been studied.

Here, we use in vivo miniature microscope Ca$^{2+}$ imaging, as well as in vivo and ex vivo optogenetic approaches and circuit tracing, to investigate the role of D1R+, and Adora+ neurons located in the ventral part of the tail striatum (vTS) in auditory-cued fear learning.

## Results

### Coding for footshocks, tones and movement by D1R+ vTS neurons

We started by imaging the activity of vTS neurons during a 3-day fear learning paradigm (*Figure 1A*). To access the vTS, a deep brain area close to the basolateral - and central nuclei of the amygdala, we used miniature-microscope Ca$^{2+}$ imaging (*Figure 1B and C*; *Ghosh et al., 2011*). In a first series of experiments, we used *Drd1a$^{Cre}$* mice, to target the expression of GCaMP6m to neurons of the direct pathway, using a Cre-dependent AAV vector (*Figure 1B and C*; Materials and methods). The *Drd1a$^{Cre}$* mouse chosen for this purpose (line EY217 from GenSAT; see Materials and methods) shows expression of Cre in the vTS, but no expression was observed in the adjacent cortical or claustrum structures (*Figure 1D*; *Figure 1—figure supplement 1*; see also *Gerfen et al., 2013*). This allowed us to target striatal neurons selectively by miniature-microscope imaging with the employed mouse line (*Figure 1—figure supplement 2*).

The fear conditioning protocol consisted of three sessions given on subsequent days (*Figure 1A*). On the first day, mice experienced a habituation session during which six 30 s CS stimulation blocks consisting of 7 kHz tone beeps were applied (see Materials and methods). During this session, only a small sub-population of D1R+ neurons (6/176) showed a response to the tone beeps (*Figure 1E, I*). One day later, each of the six CS blocks was followed by a 1 s footshock, to which 45/163 imaged

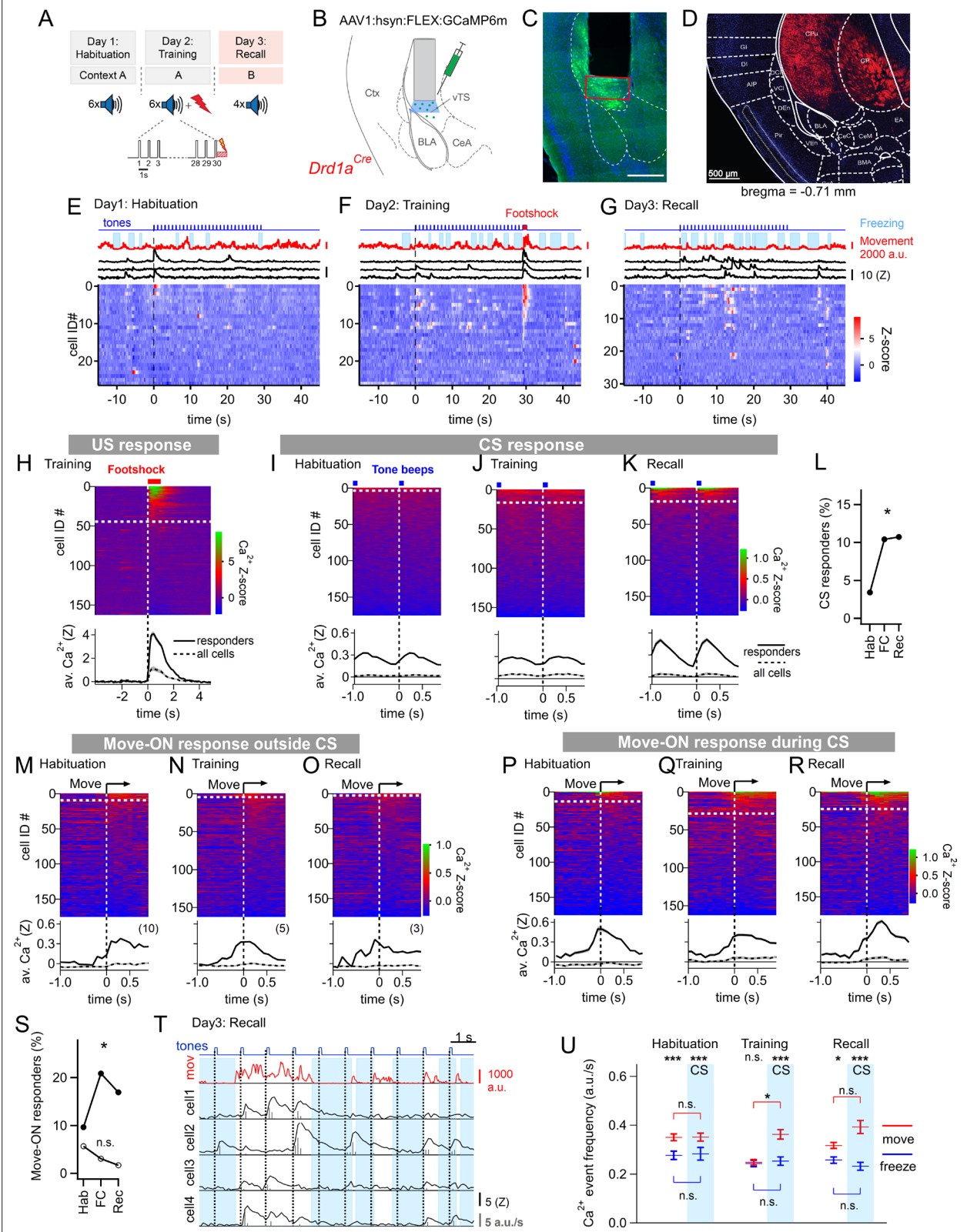

**Figure 1.** Miniature microscope Ca²⁺-imaging of D1R+ neurons in the vTS reveals coding for tones and movement during fear learning. (**A**) Outline of the fear learning protocol. (**B**) Experimental scheme of the injection of an AAV vector and placement of the GRIN lens in the vTS of *Drd1a*^Cre^ mice. (**C**) Posthoc fluorescence microscopy image from an injected *Drd1a*^Cre^ mouse expressing GCaMP6m (green channel; blue channel: DAPI). The black region indicates the position of the GRIN lens. The putative imaging area is depicted with a red rectangle. Scalebar, 500 µm. (**D**) tdTomato expression

*Figure 1 continued on next page*

*Figure 1 continued*

in *Drd1a*$^{Cre}$ x *Rosa26*$^{LSL-tdTomato}$ mice indicates localization of D1R+ neurons in the vTS, but not in neighboring cortical - nor amygdalar structures. (**E - G**) Movement traces - and freezing-state of an example mouse (red traces and light blue areas, respectively), Z-scored Ca$^{2+}$ traces for three example neurons (black traces); and color-coded Z-scored Ca$^{2+}$ traces for all neurons in one example mouse (bottom). Data are from the fourth CS presentation of day 1 (**E**), the fourth CS-US presentation of day 2 (**F**), and the second CS presentation on day 3 (**G**). (**H**) Color-coded Z-scored Ca$^{2+}$ responses to footshocks in all imaged D1R+ neurons from N=8 mice. Responses with Z>1 in a time interval of [0; 1 s] were considered as significant (see white dashed line). The bottom panel shows the average ± S.E.M. of Ca$^{2+}$ traces for all responders (n=45 neurons; black trace ± gray shades), and the average ± S.E.M. across all neurons (n=163; dashed black trace ± gray shades). (**I - K**) Z-scored Ca$^{2+}$ traces aligned to tone beeps (CS) during the habituation day, training day, and recall day (I, J, K, respectively). Responses with an average Z-score >0.2 in the time interval [0; 0.5 s] were considered significant. The black traces in the lower panel follow the same logics as in (**H**). (**L**) Percentage of tone - responsive neurons for each day. (**M - O**) Color-coded Z-scored Ca$^{2+}$ traces aligned to the movement - ON events, analyzed at times in-between CS blocks, for the habituation -, training - and recall days, as indicated (*top* panel). The traces in the bottom panel were analyzed as in (**H**). (**P - R**) Ca$^{2+}$ traces aligned to the movement - ON events, analyzed during the 30 s tone blocks (CS), for the habituation -, training - and recall days (P, Q, and R respectively). (**S**) Percentage of movement - ON responding neurons during the CS, and in the absence of a CS (closed, and open symbols, respectively). (**T**) Example traces of, from *top* to *bottom*, times of tone beeps (blue trace); movement index (red trace); and Ca$^{2+}$ traces from four example neurons in one mouse; times of freezing are highlighted by light blue. The detection of Ca$^{2+}$ - events and their amplitudes by a deconvolution analysis is indicated by vertical bars (see Materials and methods). (**U**) The amplitude - weighted frequency of Ca$^{2+}$ events (average ± S.E.M.) is plotted separately for the four combinations of CS / no CS epochs, and movement / freezing states of the mice, for the habituation, training, and fear memory recall day. The presence of a CS is indicated by the blue bars. For statistical parameters, see Results text.

The online version of this article includes the following source data and figure supplement(s) for figure 1:

**Source data 1.** Raw data and statistical tests for *Figure 1* and its supplements.

**Figure supplement 1.** Localization of Cre-expressing cells in the vTS of *Drd1a*$^{Cre}$ x *Rosa26*$^{LSL-tdTomato}$ mice.

**Figure supplement 2.** Post - hoc histology of GCaMP6m expression and GRIN lens placement in *Drd1a*$^{Cre}$ mice.

**Figure supplement 3.** Correction of CS responses for movement - ON responses.

**Figure supplement 4.** Correction of movement - ON responses for CS – responses.

**Figure supplement 5.** Individual data points and statistical comparison for the Ca$^{2+}$ event frequencies for four different conditions.

D1R+ neurons responded with robust Ca$^{2+}$ signals (*Figure 1F and H*). During this training session, an increased number of D1R+ neurons responded to tone beeps (CS) (*Figure 1J and L*; Chi-square test, p=0.018, $X^2_{df=2}$ = 8.004); the average Ca$^{2+}$ response of the neurons that responded was similar to the one observed during the habituation session (*Figure 1I and J*). During the training- and recall session, we also observed that tone beeps were in 10–15% of the cases followed by movement transitions; for the calculation of the number of CS - responders, these trials were removed (see *Figure 1—figure supplement 3*). Finally, during a fear memory recall session on day 3, the CS was presented alone in a different context. During this session, the number of tone-responsive neurons was comparable to the one during the training session (*Figure 1L*); the average Z-scored Ca$^{2+}$ signal was increased above the response amplitude on the training session (*Figure 1J and K*; 0.366±0.071 vs 0.093±0.034 for n=19 and 17 responders, respectively; 95% CI: [0.2164; 0.516]; [0.0212; 0.1649]; p=0.001, U=61, Mann-Whitney test). Thus, in vivo Ca$^{2+}$ imaging showed that a significant fraction of D1R+ neurons in the vTS responds to footshocks, and furthermore, that D1R+ neurons increase their responsiveness to the CS.

During fear learning, rodents acquire a defensive behavior in response to a CS, in the form of freezing (*LeDoux, 2000*; *Fanselow, 2018*). The freezing bouts of mice typically lasted a few seconds, and were interrupted by movement re-initiation (see e.g. *Figure 1E–G*, *top*). We asked whether transitions from freezing to movement would drive activity in D1R+ vTS neurons; we first restricted this analysis to times when no tones (CS) were presented. Aligning GCaMP6m fluorescence traces to the movement onset revealed Ca$^{2+}$ events in a sub- population of D1R+ neurons; we call these 'movement-ON' responses. The number of responding neurons decreased across the 3-day fear learning protocol, although this trend did not reach statistical significance (*Figure 1M–O*; *Figure 1S*, open symbols; p=0.117; $X^2_{df=2}$ = 4.292, Chi-square test). We next analyzed movement-ON responses during the 30 s CS presentations (*Figure 1P–R*). We found that a substantial number of movement-ON transitions were preceded by a tone; for the calculation of the percentage of movement-ON responders, these events were removed (*Figure 1—figure supplement 4*). The analysis showed that the number of D1R+ neurons that responded to a movement-ON transition during the time of the CS presentations increased during the training session, and was maintained at an elevated level during the

recall session (**Figure 1S**, filled symbols; p=0.0155, $X^2_{df=2}$ = 8.340, Chi-square test). Furthermore, the number of neurons responding to a movement-ON transition was always higher during the CS, as compared to no - CS periods (**Figure 1S**). Taken together, this data suggests that subpopulations of D1R+ neurons, in addition to responding to footshocks- and to tone stimulation, also code for movement onset.

We next analyzed more comprehensively how the Ca²⁺-event frequency depends on the movement state of the animal *and* on the presence or absence of a CS. For this, we first deconvolved the fluorescence traces to obtain times of Ca²⁺ events and their amplitudes (**Figure 1T**; Materials and methods; *Pnevmatikakis et al., 2016*; *Giovannucci et al., 2019*). This allowed us to compute the amplitude-weighted frequency of Ca²⁺ events during the four combinations of movement states / tone presentations, for each day of the fear learning protocol (**Figure 1U**; **Figure 1—figure supplement 5**). The data was significantly different across conditions (p<0.0001, KW = 138.33, Kruskal-Wallis test). Pairwise comparisons of these conditional Ca²⁺ event frequencies within each session showed that on most sessions, the activity of D1R+ neurons was significantly higher when the mice moved, than when they froze, with the exception of the no-CS times on the training day (**Figure 1U**; Dunn's multiple

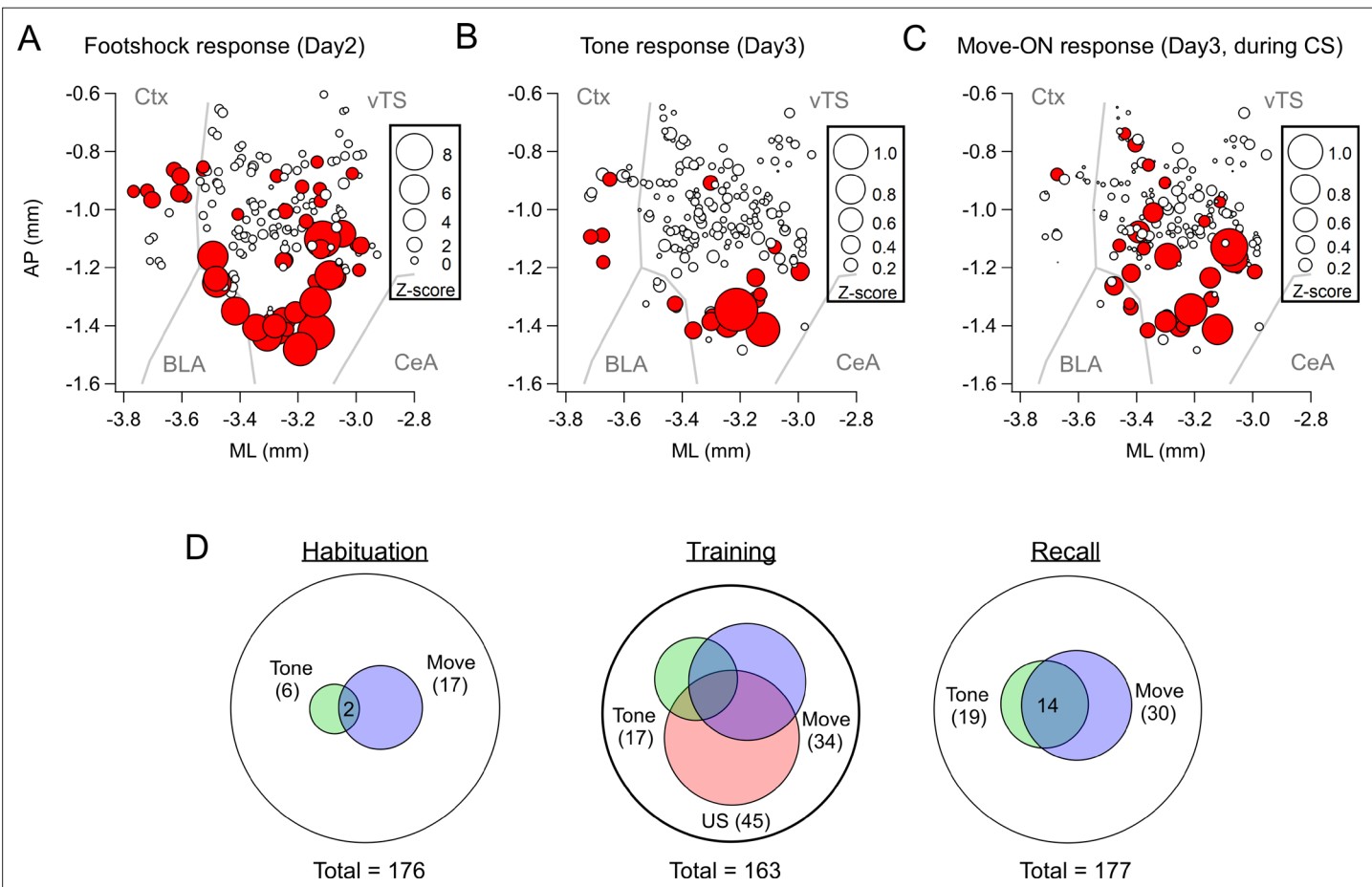

**Figure 2.** Spatial localization of the imaged D1R+ vTS neurons, and overlap of neurons coding for sensory events and movement state. (**A–C**) Maps with the position of all imaged D1R+ vTS neurons, plotted as projection on the horizontal plane. Shown are the footshock-responses (**A**), the tone (CS) responses as imaged on day3 (**B**), and the movement-onset responses imaged on day 3 (**C**). Neurons are drawn in red when their response was considered significant (average Z-score >1 for footshock responses; average Z-score >0.2 for tone, and movement - ON responses; note different scales of the circles). For a coarse orientation, outlines of the amygdalar nuclei (basolateral amygdala; 'BLA' and central amygdala, 'CeA') and cortex ('Ctx') are shown. (**D**) Venn diagrams showing the overlap of neuronal populations within the D1R+ vTS neurons that respond to tones (green), to movement-ON events (blue), and to footshock stimulation (red; on day 2 only). Note the increased number of neurons responding to tones and movement - onset during fear memory recall, and the overlap of these sub-populations (right panel).

The online version of this article includes the following figure supplement(s) for figure 2:

**Figure supplement 1.** Maps with the position of the imaged D1R+ neurons, plotted as a projection onto the coronal plane.

comparisons test; for p-values, see legend to *Figure 1—figure supplement 5*). On the training day, given that mice moved, the $Ca^{2+}$ event frequency was significantly increased by the CS (*Figure 1U*; p=0.0163, Mean rank difference = 241.9, Kruskal-Wallis test followed by Dunn's multiple comparisons test). Thus, in vivo $Ca^{2+}$ imaging during a 3-day fear learning protocol shows that the activity of the D1R+ neurons in the vTS is higher during movement than during the freezing state of the mice. Moreover, fear learning increases the number of neurons with a phasic response to tones (CS), as well as the number of neurons with a movement-ON response during the CS presentation (*Figure 1L and S*).

To determine the location of the imaged neurons and to compare them across mice, we aligned the center of the GRIN lens to a mouse brain atlas in each mouse, and generated a common cell map based on the cell coordinates relative to the lens center (Materials and methods). This revealed a hotspot of footshock- and tone-responding D1R+ neurons in the posterior-ventral region of the tail striatum medial to the LA (*Figure 2A and B*; *Figure 2—figure supplement 1*). D1R+ neurons with movement-ON responses during the recall sessions were located in a similar area (*Figure 2C*). Venn plots of the overlay of the various response types showed that during the habituation session, neurons with responses to the CS and movement did not strongly overlap, and represented a small proportion of the imaged neurons (*Figure 2D*). During the training session, the neurons with tone- and movement-ON responses increased in numbers (see also *Figure 1L and S*), and about half of each sub-population also showed a footshock response. During the recall session, the populations of both tone - and movement responders stayed constant with respect to the training day, and these response types now overlapped substantially (*Figure 2D*). Taken together, in vivo $Ca^{2+}$ imaging shows that a subpopulation of D1R+ vTS neurons, located in a posterior-ventral hotspot of the tail striatum, responds to footshocks. During the course of fear learning, these neurons increasingly code for an aversively motivated CS and for movement-ON transitions, suggesting that these representations in populations of D1R+ vTS neurons undergo plasticity driven by fear learning.

## Coding for footshocks and movement by Adora+ vTS neurons

The other large population of principal neurons in the vTS are Adora+ neurons which, in analogy to other striatal areas, represent neurons of an 'indirect' pathway through the basal ganglia (*Gerfen et al., 1990*; *Gangarossa et al., 2019*). We next investigated the in-vivo activity of this population of vTS neurons throughout the three-day fear learning protocol, using an *Adora2a^Cre* mouse line to target the expression of GCaMP6m to Adora+ neurons in the vTS (*Figure 3A*; *Figure 3—figure supplements 1 and 2*; Materials and methods). About forty percent of the Adora+ vTS neurons responded to footshocks presented during the training session (79/201 neurons; *Figure 3C and E*). A small subpopulation (13/173 or ~8%) responded to tone beeps during the CS (*Figure 3F*). Contrasting with the D1R+ neurons, the percentage of tone (CS)-responsive neurons did not change during fear learning (*Figure 3F–H*; *Figure 3I*; p=0.415, $X^2_{df=2}$ = 1.756, Chi-square test). Thus, a large sub-population of Adora+ neurons in the vTS responds to footshocks, but the number of Adora+ neurons that responds to tones (CS) remains unchanged during fear learning.

We next analyzed whether the activity of Adora+ neurons in the vTS was modulated by the movement state of the mice. In the absence of tones, a moderate sub-population of Adora+ neurons showed movement-ON responses during the habituation session (27/173 or ~16 %); this number decreased over the course of fear learning (*Figure 3J–L*; *Figure 3P*, open symbols; p=0.0229, $X^2_{df=2}$ = 7.557; Chi-square test). In contrast, during the CS, there was a larger number of Adora+ neurons that showed movement-ON responses; their number was unchanged over the three-day fear learning protocol (*Figure 3M–O*; *Figure 3P*, closed symbols; p=0.717, $X^2_{df=2}$ = 0.67; Chi-square test). These experiments show that a substantial sub-population of Adora+ neurons in the vTS codes for movement onset, but this representation was unchanged by fear learning (*Figure 3M–P*), except for a decrease in the number of neurons showing a movement - ON response in the absence of a CS (*Figure 3P*, open symbols).

Similarly as for the D1R+ neurons, we next analyzed the activity of Adora+ neurons as a function of the four combinations of movement state (movement versus freezing) and CS presentation (presence, or absence of a CS) (*Figure 3Q and R*). A Kruskal-Wallis test showed that the amplitude-weighted frequency of $Ca^{2+}$ events differed across categories (*Figure 3R*; p<0.0001, KW = 184.6). During all three behavior sessions, and irrespective of the presence or absence of a CS, the activity of Adora+ neurons was larger during movement than during freezing (*Figure 3R*; see legend to *Figure 3—figure*

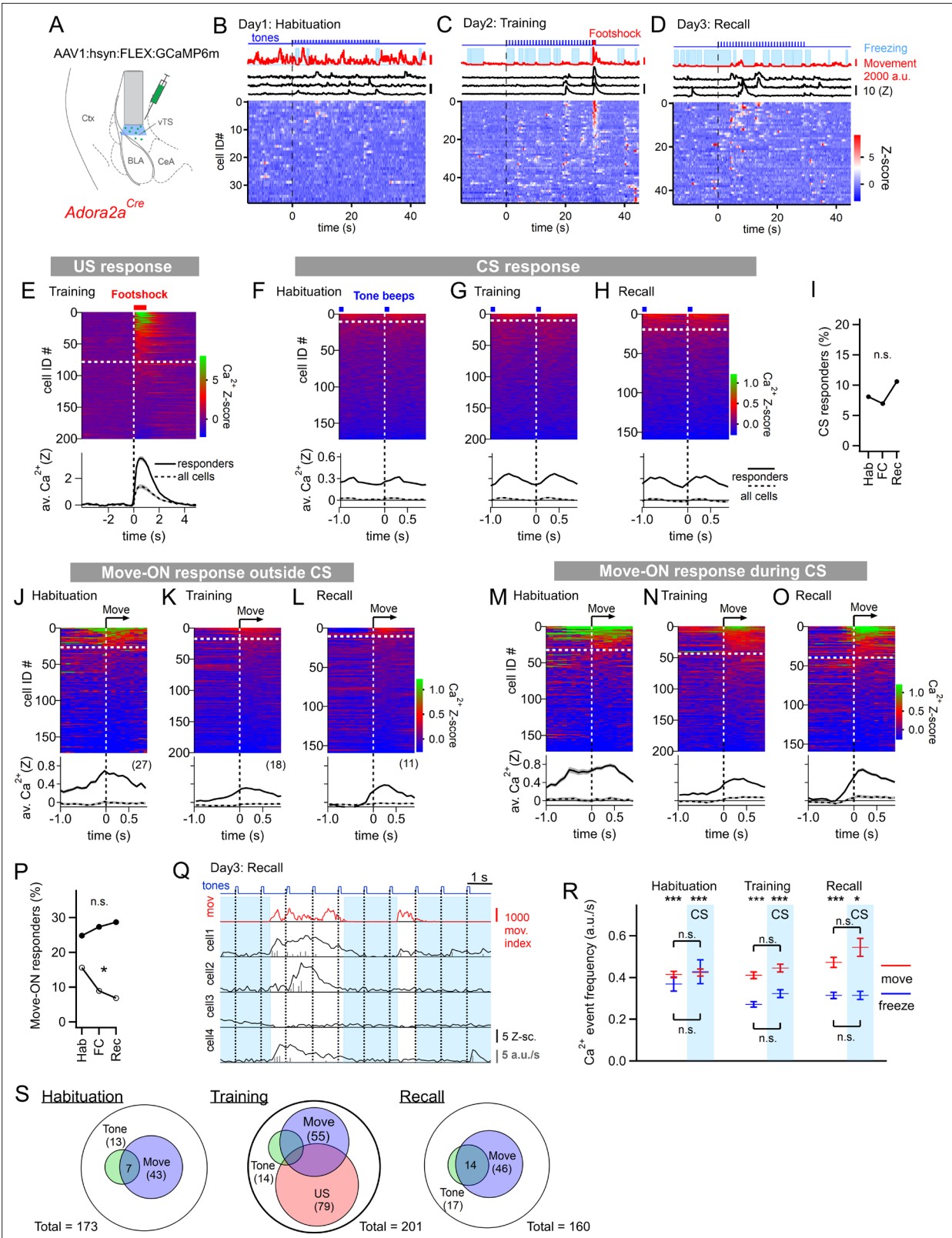

**Figure 3.** Adora+ neurons in the vTS code for footshocks and movement onset, but CS coding is less present. (**A**) Experimental scheme of the injection of an AAV vector and placement of the GRIN lens in *Adora2a^Cre* mice. (**B–D**) Movement traces - and freezing-state of an example mouse (red traces and light blue areas, respectively), Z-scored Ca²⁺ traces for three example neurons (black traces); and color-coded Z-scored Ca²⁺ traces for all neurons in one example mouse (bottom). Data are from the second CS presentation of day 1 (**D**), the fourth CS-US presentation of day 2 (**E**), and the second

*Figure 3 continued*

CS presentation on day 3 (**F**). (**E**) Z-scored Ca²⁺ responses to footshocks in all imaged Adora+ neurons from N=8 mice; responses with average Z>1 in the interval of [0; 1 s] were considered as significant (traces above dashed white line). The bottom panel shows the average ± S.E.M. of Ca²⁺ traces for all responders (n=79 neurons; black trace ± gray shades), and the average ± S.E.M. across all neurons (n=201; dashed black trace ± gray shades). (**F – H**). Color-coded Z-scored Ca²⁺ traces aligned to tone beeps (CS) during the habituation day, training day, and recall day (F, G, H, respectively). Responses with average Z-score >0.2 in the time interval of [0; 0.5 s] were considered significant; the black traces in the lower panel were calculated as in (**E**). (**I**) Percentage of tone-responsive neurons for each day. (**J - L**). Ca²⁺ responses to movement-ON events, analyzed outside the CS blocks, for the habituation-, training-, and recall days (J, K, and L, respectively). The top and bottom panels follow the same logics as in (**E**). (**M - O**) Ca²⁺ responses to movement-ON events, analyzed during the 30 s tone blocks (CS), for the habituation, training, and recall days as indicated. (**P**) Percentage of movement - ON responders in the presence and absence of a CS (closed, and open symbols). (**Q**) Illustration of the Ca²⁺ deconvolution approach for four example neurons in a *Adora2a*^*Cre*^ mouse. From *top* to *bottom*, times of tone beeps (blue trace); movement index (red trace); and Ca²⁺ traces from four example neurons; times of freezing are highlighted by light blue. Vertical gray bars indicate the timing and amplitude of the detected Ca²⁺ events. (**R**) The amplitude-weighted frequency of Ca²⁺ events (average ± S.E.M.), analyzed separately for the four combinations of CS / no CS times, and movement / freezing states of the mice, for the 3 fear learning days. The presence of a CS is indicated by the blue bars. The p-values for the indicated statistical comparisons are reported in the Results text. (**S**) Venn diagrams showing the overlap of neuronal populations within the Adora+ vTS neurons that respond to tones (green), to movement-onset transitions (blue), and to footshock stimulation.

The online version of this article includes the following source data and figure supplement(s) for figure 3:

**Source data 1.** Raw data and statistical tests for *Figure 3* and its supplements.

**Figure supplement 1.** Localization of Cre-expressing cells in the vTS of *Adora2a*^*Cre*^ x *Rosa26*^*LSL-tdTomato*^ mice.

**Figure supplement 2.** Post-hoc histology of GCaMP6m expression and GRIN lens placement in *Adora2a*^*Cre*^ mice.

**Figure supplement 3.** Individual data points and statistical comparison for the Ca²⁺ event frequencies for four different conditions.

**Figure supplement 4.** Maps with the position and Ca²⁺ signal intensity of the imaged Adora+ vTS neurons.

*supplement 3* for the corresponding p-values; post-hoc Dunn's multiple comparison test). On the other hand, the presence or absence of a CS did not significantly modulate the activity of Adora+ neurons, irrespective of whether the mice moved, or froze (*Figure 3R*; see legend to *Figure 3—figure supplement 3* for p-values; post-hoc Dunn's multiple comparison test). This analysis thus corroborates our finding that the activity of Adora+ neurons is only little modulated by tones and that it is more strongly modulated by movement, but that neither of the two representations are modulated in a plastic fashion by fear learning. The spatial distribution of Adora+ neurons responding to footshocks, tones and movement-ON transitions was overall similar to the one of D1R+ neurons (*Figure 3—figure supplement 4*). Taken together, in vivo Ca²⁺ imaging shows that a substantial percentage of Adora+ neurons in the vTS responds to footshocks, and to movement-ON transitions, and a smaller sub-population of these neurons responds to tones (*Figure 3S*). However, Adora+ neurons do not change their responses to tones and movement-ON transitions during fear learning.

## D1R+ and Adora+ vTS neurons do not instruct freezing or movement

In vivo Ca²⁺ imaging showed that sub-populations of neurons within the two main types of principal neurons in the vTS code for footshocks, for the CS and for movement-ON transitions; furthermore, D1R+ neurons increased their representation of tones and movements with fear learning (*Figures 1–3*). We next wished to investigate how D1R+ and Adora+ vTS neurons might contribute to fear learning. A classical model of basal ganglia function postulates that D1R+ neurons in the direct pathway initiate movement, whereas Adora+ (or D2R+) neurons in the indirect pathway suppress movements (*Kravitz et al., 2010*; *Nelson and Kreitzer, 2014*; but see *Klaus et al., 2019*). Thus, one possible straightforward hypothesis is that activity of Adora+ neurons of the vTS instructs an arrest of movement, or freezing, and vice-versa, that D1R+ neurons instruct movement re-initiation. We wished to test this hypothesis by optogenetic activation of either D1R+ - or Adora+ vTS neurons, at times when naive mice (that had not undergone fear learning) are engaged in regular exploratory behavior. For this, the channelrhodopsin variant Chronos (*Klapoetke et al., 2014*) was expressed bi-laterally and Cre-dependently in D1R+, or Adora+ neurons of the vTS, using AAV1:hSyn:FLEX:Chronos-eGFP in the respective Cre-mouse line (*Figure 4A*; *Figure 4—figure supplements 1 and 2*). Three to 4 weeks later, the mice were allowed to explore the fear conditioning chamber, and trains of optogenetic stimuli were applied at pre-determined intervals, irrespective of whether the mice moved or paused from movement (pulse duration 1ms, repeated at 25 Hz for 2 s; each train given six times for each mouse). Optogenetic stimulation did not lead to changes in the movement activity of the

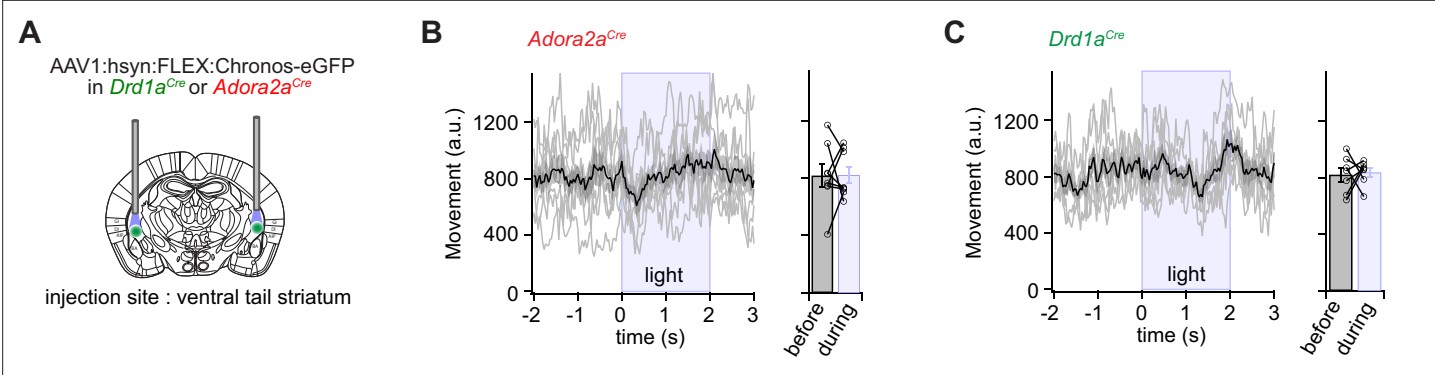

**Figure 4.** Optogenetic stimulation of D1R+ and Adora+ vTS neurons has no direct effect on movement. (**A**) Scheme showing the bilateral placement of optic fibers over each vTS, and the injection of an AAV vector driving the Cre-dependent expression of Chronos. (**B**) *Left*, *Adora2aCre* mice expressing Chronos in Adora+ neurons of the vTS were allowed to explore the fear conditioning chamber, and six trains of blue light stimuli (50 pulses of 1ms length, 25 Hz) were applied. The average movement indices for N=8 mice are shown, centered around the time of light stimulation. *Right*, individual and average movement data for N=8 mice, for 2 s intervals before and during optogenetic stimulation. (**C**) Analogous experiment to the one in (**B**), now performed for *Drd1aCre* mice expressing Chronos in a Cre-dependent manner in the vTS (N=7 mice). Note that optogenetic stimulation of neither Adora+ neurons (**B**), nor of D1R+ neurons led to notable changes in the movement of mice (see Results for the statistical parameters).

The online version of this article includes the following source data and figure supplement(s) for figure 4:

**Source data 1.** Raw data and statistical tests for *Figure 4* and its supplements.

**Figure supplement 1.** Post-hoc histology of Chronos expression and optic fiber placement in *Adora2aCre* mice.

**Figure supplement 2.** Post-hoc histology of Chronos expression and optic fiber placement in *Drd1aCre* mice.

**Figure supplement 3.** Additional optogenetic stimulation experiments with longer pulse widths.

**Figure supplement 4.** Post - hoc histology of Chronos expression and optic fiber placement, for the additional experiments with *Adora2aCre* mice.

*Adora2aCre* mice, nor of the *Drd1aCre* mice (*Figure 4B and C*; N=8 and 7 mice; p>0.999, W=0, and p=0.837, $t_6$=0.215, Wilcoxon and paired t-test, respectively). In additional experiments with Adora+ neurons, we employed longer light pulses (2 and 5 ms), but we similarly did not observe effects on the movement state of the mice (*Figure 4—figure supplements 3 and 4*). These experiments suggest that in naive mice, the activity of neither D1R+ - nor of Adora+ neurons is sufficient to modulate the movement activity of the mice.

## Footshock-driven activity of D1R+ vTS neurons contributes to fear learning

We found that many D1R+ neurons respond to footshocks, and that following fear learning, these neurons show an increased response to the CS and to movement-ON transitions (*Figures 1 and 2*). This suggests that footshock responses might drive a plasticity of CS - representation, and of the movement-ON representation in D1R+ vTS neurons. To investigate the role of the footshock-evoked activity of D1R+ vTS neurons in fear learning, we next silenced the activity of these neurons during footshock presentation, and observed the effects of this manipulation on freezing behavior during the training day, and one day later during fear memory recall.

For optogenetic silencing, we expressed the light-sensitive proton pump Archaerhodopsin (Arch; *Chow et al., 2010*) in a Cre-dependent manner bilaterally in the vTS of *Drd1aCre* mice, and implanted optic fibers over each vTS (*Figure 5A*; *Figure 5—figure supplement 1*). Mice in a control group were injected with an AAV vector driving the expression of eGFP (Materials and methods). Four weeks later, both groups of mice underwent auditory-cued fear learning, and on the training day yellow laser light (561 nm) was applied for 3 s, starting 1 s before each footshock, with the aim to suppress the footshock-driven activity of D1R+ neurons (*Figure 5B*). We found in ex vivo experiments that yellow light strongly hyperpolarizes Arch-expressing D1R+ and Adora+ vTS neurons, suppresses action potential (AP) firing, and does not evoke rebound APs when the light is switched off (*Figure 5—figure supplement 2*).

During the habituation day of the fear learning protocol the mice showed little freezing, as expected (*Figure 5C*). On the training day, during which mice received a 1 s footshock after each tone block,

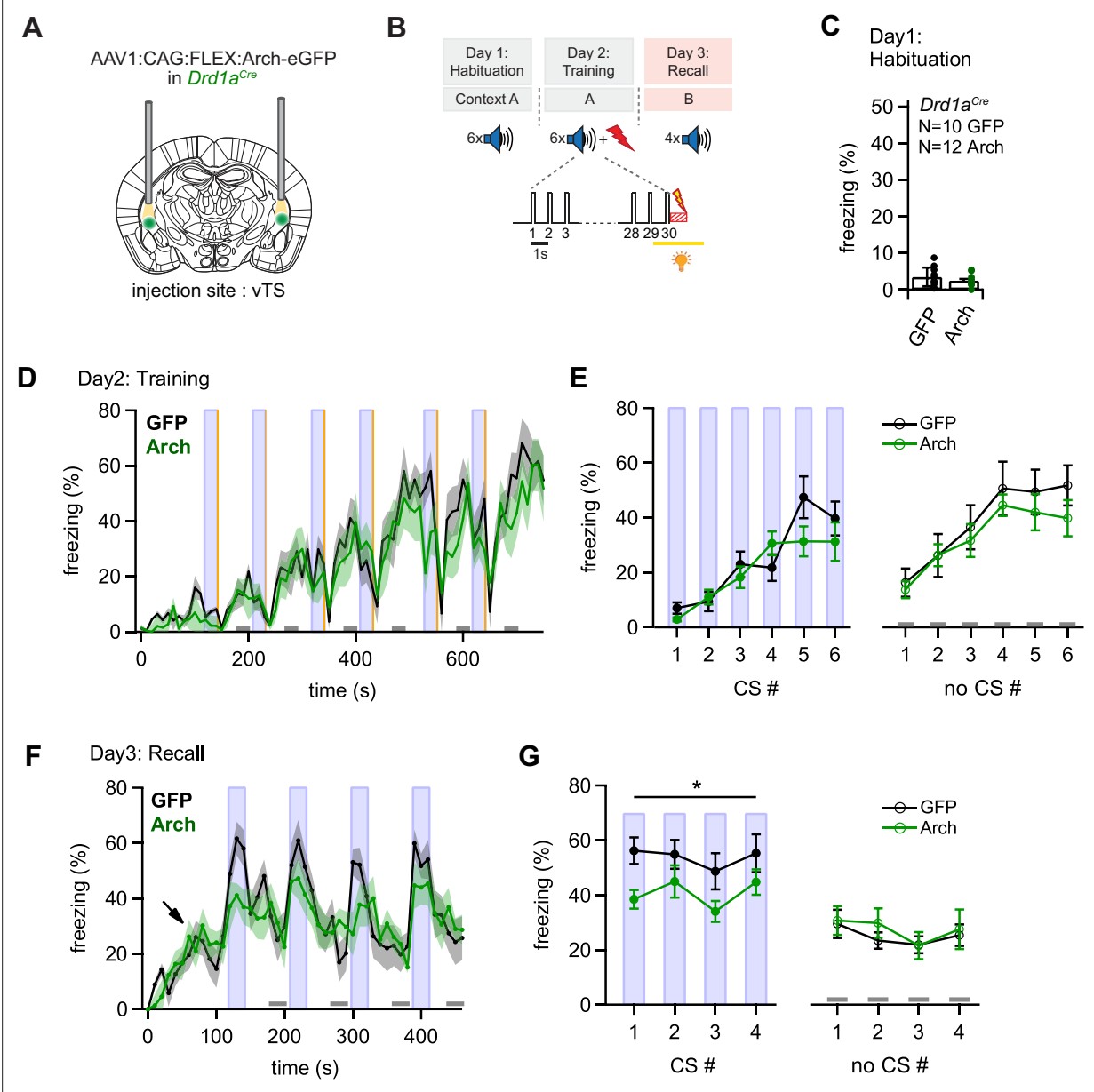

**Figure 5.** Optogenetic inhibition of D1R+ vTS neurons during footshock reduces the formation of an auditory-cued fear memory. (**A**) Scheme showing the bi-lateral injection in *Drd1a*^Cre mice of an AAV vector driving the expression of either Arch or eGFP (for controls), and the placement of optic fibers over each vTS. (**B**) Behavioral paradigm outlining the 3-day fear learning protocol, and the application of yellow light during the footshock presentation on the training day. (**C**) Freezing levels during 30 s CS presentations on the habituation day for the control mice (expressing eGFP; black circles), and for Arch-expressing mice (green circles). (**D**) Time-binned average percent freezing traces for control mice (black data) and Arch-expressing mice (green data; bin width is 10 s in both cases). Light blue boxes indicate the 30 s CS presentation periods. Gray bars (bottom) indicate epochs of 'no CS' analysis. Vertical orange lines indicate the times of footshock presentation. (**E**) Quantification of freezing during the training day for the control mice and Arch-expressing mice (black and green data, respectively). Freezing was averaged during the 30 s CS presentations (filled circles, *left*) and for the no CS epochs (open circles, *right*). (**F**) Time-resolved freezing during the cued retrieval on day 3. Note the increases in freezing driven by each CS presentation in control mice (black trace, average of N=10 eGFP expressing mice), which were smaller in amplitude in Arch-expressing mice (N=12, green trace). Light blue areas and gray bars (bottom) indicate times of CS presentation, and epochs of 'no CS' analysis, respectively. (**G**) Quantification of freezing during the 30 s CS presentation (*left*) and during the no CS epochs (*right*). Note the significant reduction of freezing during the CS in the Arch group (see Results for statistical parameters).

The online version of this article includes the following source data and figure supplement(s) for figure 5:

**Source data 1.** Raw data and statistical tests for *Figure 5* and its supplements.

**Figure supplement 1.** Post-hoc histology of Arch expression and optic fiber placement in *Drd1a*^Cre mice.

**Figure supplement 2.** Properties of light-activated archaerhodopsin in D1R+ - and Adora+ vTS neurons.

mice in both groups showed increasing levels of freezing throughout the training period, interrupted by low freezing activity immediately following the footshocks, caused by increased shock-evoked running and escape behavior (**Figure 5D**; CS periods and footshocks are indicated by blue and yellow vertical lines, respectively). To quantify freezing behavior, we averaged the percent freezing during the CS, and during six no-CS epochs (see **Figure 5D**, light blue vertical lines, and lower gray bars, respectively). This showed that freezing levels were not different between the Arch- and the control group, neither for the CS- nor for the no-CS epochs (**Figure 5E**, *left*, for the CS epochs: p=0.338, $F_{1,20}$ = 0.965; **Figure 5E** *right*, for the no-CS epochs: p=0.454, $F_{1,20}$=0.583; two-way repeated measures - ANOVA). Thus, optogenetic silencing of the footshock-evoked activity of D1R+ vTS neurons did not change the freezing behavior of mice during the training day.

On the third day, we tested for fear memory recall by applying tone stimulation (CS) alone in a different context (see Materials and methods). The time-resolved freezing analysis revealed a gradual increase of freezing levels when mice entered the conditioning chamber (to ~20%; **Figure 5F**, arrow). This baseline level of freezing has been observed before (see e.g. **Cummings and Clem, 2020**), and likely represents a residual contextual fear memory, despite the change of the context between the training day and the recall day. The 30 s tone blocks (CS) caused a vigorous increase in freezing in eGFP-expressing control mice, whereas in the Arch-expressing mice, the CS was less efficient in driving freezing (**Figure 5F**, black- and green average traces, N=10 eGFP- and N=12 Arch-expressing $Drd1a^{Cre}$ mice). Averaging and statistical analysis revealed a significant difference in CS-driven freezing between control - and Arch-expressing $Drd1a^{Cre}$ mice (**Figure 5G**, closed data points; p=0.040, $F_{1,20}$=4.853, two-way repeated measures - ANOVA). On the other hand, freezing during the no-CS epochs was unchanged between the two groups (**Figure 5G**, open data points; p=0.662, $F_{1,20}$=0.197 two-way repeated measures - ANOVA). Thus, optogenetic inhibition of D1R+ neurons in the vTS during the footshocks on the training day causes a diminished auditory-cued recall of fear memory 1 day later. These data suggest that footshock-driven activity of D1R+ vTS neurons contributes to auditory-cued fear learning.

## Adora+ vTS neurons suppress learned fear in the absence of a CS

We next investigated the role of footshock-driven activity in the Adora+ vTS neurons for auditory-cued fear learning. For this, we silenced the activity of Adora+ vTS neurons during the footshocks presented on the training day (**Figure 6A and B**; **Figure 6—figure supplement 1**), in an approach analogous to the one used for the D1R+ neurons. During the habituation day, we observed low freezing as expected (**Figure 6C**). On the training day, mice in both the eGFP-expressing control group and in the Arch group showed a gradual increase in freezing with successive CS-US pairings, interrupted only by low freezing activity immediately following the footshocks, as in the $Drd1a^{Cre}$ mice (**Figure 6D**). The analysis of freezing during the CS- and no-CS epochs revealed no differences in the freezing levels between the Arch- and the eGFP groups on the training day (**Figure 6E**; p=0.528, $F_{1, 17}$ = 0.415 and p=0.312, $F_{1,17}$ = 1.087, respectively; two-way repeated measures ANOVA). Thus, similar as for the D1R+ neurons, footshock-driven activity in Adora+ vTS neurons is not necessary for the freezing behavior that develops during the training day.

On the fear memory recall day, the dynamics of the freezing behavior differed between the eGFP and the Arch group (**Figure 6F and G**). While the eGFP-expressing control mice displayed an increased freezing during each CS epoch followed by a relaxation to lower freezing levels, mice in the Arch group showed a delayed relaxation of freezing following the CS epochs (**Figure 6F**; black and red data, respectively). To quantify these effects, we analyzed the average time spent freezing during the CS and during a late no-CS epoch (**Figure 6F**, light blue bars, and lower gray bars). This showed a trend towards an *increased* freezing during the no-CS epochs in the Arch group as compared to eGFP controls, although this difference did not reach statistical significance (**Figure 6G**, right; p=0.0512, $F_{1,17}$ = 4.40; two-way repeated measures ANOVA, N=8 and 11 eGFP and Arch mice, respectively). On the other hand, freezing during the CS epochs was unchanged between the Arch and the control group (**Figure 6G**, *left*; p=0.624, $F_{1,17}$ = 0.249; two-way repeated measures ANOVA). Thus, silencing the footshock-driven activity of Adora+ neurons in the vTS induces a trend towards higher freezing during fear memory recall in the *absence* of the CS. These findings, together with the results obtained from silencing D1R+ vTS neurons (see above), suggest that direct and indirect pathway neurons of the vTS have separate, but functionally synergistic roles in fear learning. In fact, the action of both

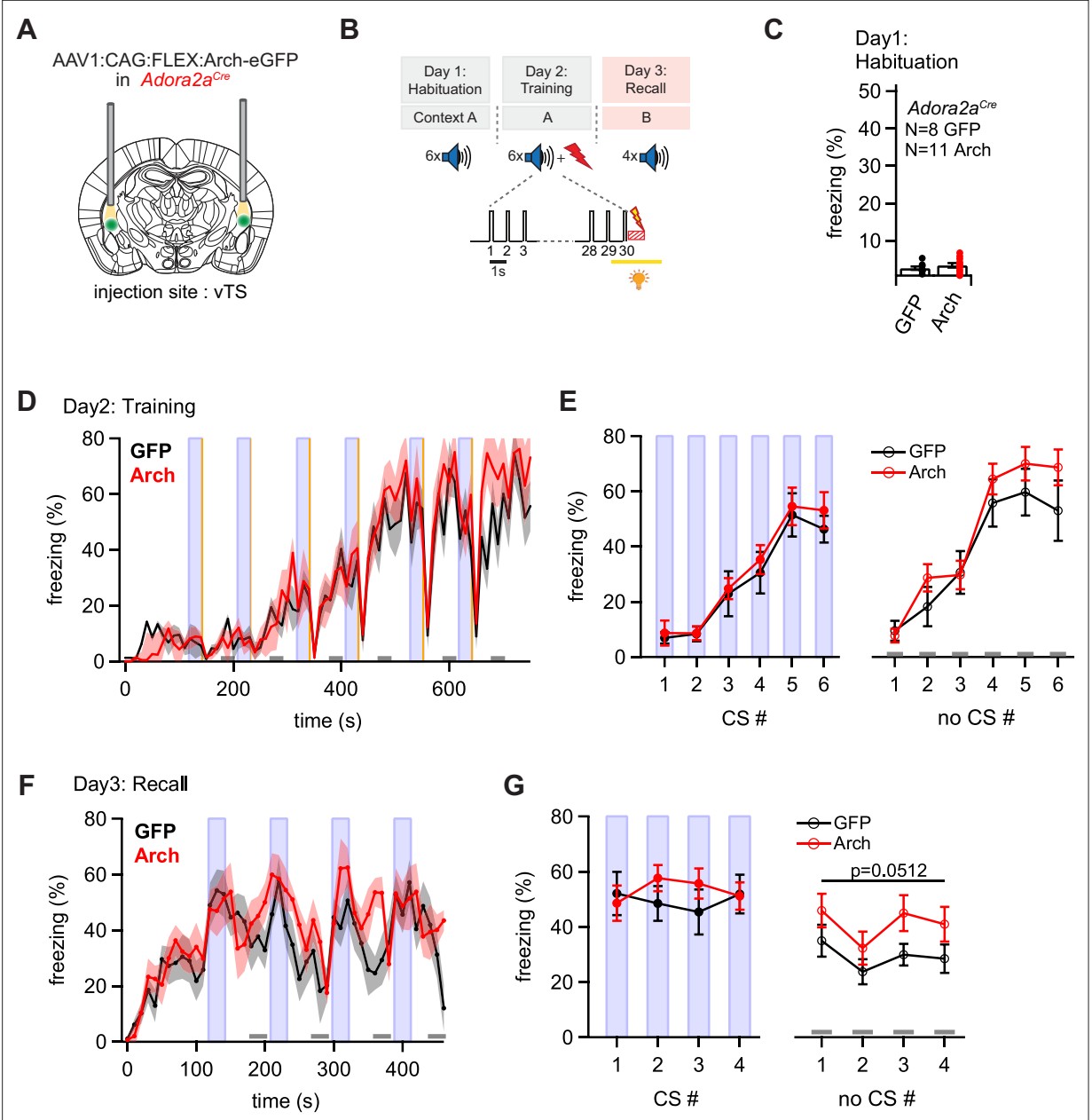

**Figure 6.** Optogenetic inhibition reveals a role of Adora+ vTS neurons in suppressing freezing in the absence of a learned CS. (**A**) Scheme of the experimental approach, during which Arch (or eGFP, for controls) was expressed Cre-dependently in *Adora2a^Cre* mice in the vTS, and optic fibers were placed in the vTS above the injection sites. (**B**) Behavioral paradigm (upper panel), and scheme of the application of yellow light during the footshock stimulus (US) on the training day (lower panel). (**C**) Freezing levels during CS presentations on the habituation day for mice expressing eGFP (control group; black data) and for Arch-expressing mice (red data). (**D**) Time-resolved analysis of freezing for control mice (black trace, average of N=8 mice) and Arch-expressing mice (red trace, N=11 mice). The light blue areas and gray bars (bottom) indicate the time of CS presentation, and the time windows for 'no CS' analysis. (**E**) Quantification of freezing during the CS (left), and during the 30 s 'no CS' epochs (right), for control mice (black) and for Arch-expressing mice (red). (**F**) Time-resolved freezing during the fear memory recall day. The light blue areas and gray bars (bottom) indicate the time of CS presentation, and the analysis window for 'no CS' analysis. (**G**) Average percent of time spent freezing, analyzed during the 30 s CS presentations (left), and during the no CS epochs (right) of the fear memory retrieval day, for both groups of mice. There was a trend towards an enhanced freezing in the Arch group at times when no CS was present (right; p=0.0512; two-way repeated measures ANOVA; see Results for further statistical parameters).

The online version of this article includes the following source data and figure supplement(s) for figure 6:

**Source data 1.** Raw data and statistical tests for *Figure 6*.

**Figure supplement 1.** Post-hoc histology of Arch expression and optic fiber placement in *Adora2a^Cre* mice.

sub-systems together increases the difference between the strength of a defensive behavior in the presence, and absence of a learned sensory cue.

## Brain-wide screening of presynaptic inputs to the vTS

Our in-vivo Ca²⁺ imaging and optogenetic experiments have revealed a differential role of D1R+ - and Adora+ vTS neurons in auditory-cued fear learning. To start investigating the role of synaptic afferents to vTS neurons in these plasticity processes, we next used monosynaptic retrograde rabies virus tracing to identify the presynaptic neuron pools that provide input to D1R+ and Adora+ neurons in the vTS (*Figure 7A*; *Wickersham et al., 2007*; *Wall et al., 2013*).

*Drd1a^Cre*, or *Adora2a^Cre* mice were injected into the vTS with an AAV-helper virus driving the Cre-dependent expression of TVA, eGFP and oG, to render the infected neurons competent for later EnvA-pseudotyped rabies virus uptake (*Figure 7A*, see Materials and methods for the specific viruses). Three weeks later, a pseudotyped delta-G rabies vector driving the expression of dsRed was injected at the same coordinates; control experiments confirmed the specificity of the Cre-dependent expression of TVA-expressing vector, and the absence of ectopic expression of the rabies vector (*Figure 7—figure supplement 1*). In this way, starter cells at the injection site in the vTS could be identified by GFP- and dsRed co-labeling (red) (*Figure 7B*, cells appearing yellow in the overlay). Trans-synaptically labeled neurons outside of the striatum were analyzed based on their expression of dsRed. We placed all presynaptic neurons found in N=2 *Drd1a^Cre* and N=2 *Adora2a^Cre* mice into brain-wide models, with cells from the two mice arbitrarily positioned on different brain sides (*Figure 7E and F*; red and blue dots, respectively). In both *Drd1a^Cre* and *Adora2a^Cre* mice, we found the highest number and density of backlabeled cells in the secondary somatosensory cortex (S2), and in the dorsal (granular and dysgranular) part of the insular cortex (InsCx; see also *Figure 7—source data 2* for a list of all brain structures with detected presynaptic neurons; *Figure 7C, D, G and H*). A sizeable number of back-labelled neurons was also found in various areas of the primary somatosensory cortex (S1), in the globus pallidus externa (GPe), and in the thalamic parafascicular nucleus (PF; *Figure 7C, D, G and H*; *Figure 7—figure supplement 2*). The GPe and the PF stood out by having high densities of presynaptic neurons to both D1R- and Adora-MSNs, even though their absolute numbers were not large (*Figure 7G and H*; black bars). Although the brain-wide distribution of presynaptic input neurons to D1R+ and Adora+ neurons in the vTS was overall similar, we detected differences on a finer scale. Thus, we found that structures known to process auditory- and multimodal sensory information, like the auditory cortex (AUD), the temporal association cortex (TeA), the posterior triangular thalamic nucleus (PoT), the posterior intralaminar thalamic nucleus (PiL) and the basolateral amygdala (BLA), were more strongly back-labeled in *Adora2a^Cre* mice than in *Drd1a^Cre* mice (*Figure 7G and H*; *Figure 7—figure supplement 3*; *LeDoux, 2000*; *Weinberger, 2007*; *Sacco and Sacchetti, 2010*; *Dalmay et al., 2019*; *Barsy et al., 2020*). Finally, we analyzed the a-p distribution of neurons in the InsCx and S2 that provide input to both D1R+ - and Adora+ neurons in the vTS (*Figure 7I and J*). Taken together, rabies-virus mediated circuit tracing shows that the vTS receives its main cortical input from the S2 and the InsCx, followed by primary somatosensory areas. Thalamic areas like PF and VPM, and basal ganglia like GPe, as well as limbic areas like the CeA and BLA especially for the Adora+ neurons, also contain sizeable numbers of neurons that provide input to the vTS.

## The posterior insular cortex provides strong excitatory drive to the vTS

Retrograde rabies-virus labeling showed that the InsCx, and the adjacent S2 are the primary cortical input areas to the vTS (*Figure 7*). We next wished to functionally validate the connection from the InsCx to the vTS, using optogenetically assisted circuit-mapping (*Petreanu et al., 2007*; *Little and Carter, 2013*; *Gjoni et al., 2018*). For this, we focussed on the connection from the posterior InsCx (pInsCx) to the vTS, and injected a viral vector driving the expression of Chronos (*Klapoetke et al., 2014*), primarily targeting the pInsCx (*Figure 8A and B*; AAV8:hSyn:Chronos-eGFP, note that some spill-over of virus into the neighboring S2 cannot be excluded). We used *Drd1a^Cre* x *Rosa26^LSL-tdTomato*, or *Adora2a^Cre* x *Rosa26^LSL-tdTomato* mice, to identify each type of principal neuron by its tdTomato fluorescence in recordings 3 to 6 weeks later. Blue light pulses (1ms) at maximal light intensity evoked robust optogenetically-evoked EPSCs of 8.8±1.6 nA in D1R+ vTS neurons (95% CI: [5.28 nA; 12.2 nA]; n=12 cells), and of 2.5±0.5 nA in Adora+ vTS neurons (95% CI: [1.49 nA; 3.58 nA], n=19 cells); the EPSC amplitudes were significantly different between the two types of principal neurons (p=0.002,

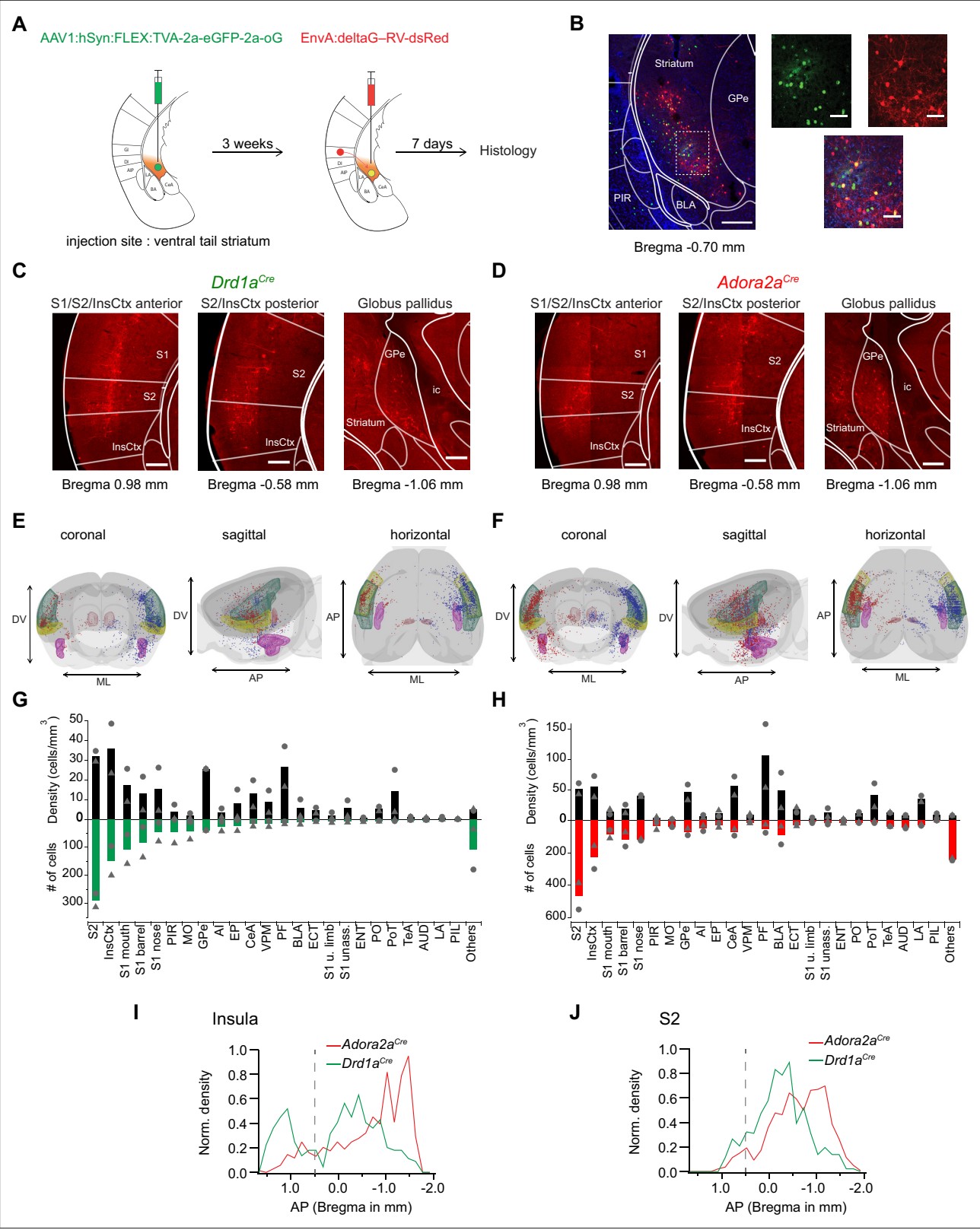

**Figure 7.** Retrograde transsynaptic tracing of brain-wide inputs to D1R+ and Adora+ vTS neurons. (**A**) Scheme of the experimental protocol for rabies-virus mediated transsynaptic tracing. (**B**) *Left*, Example confocal images of the injection site in a *Drd1a^Cre^* mouse, *Right*, confocal images at a higher magnification taken from the boxed area of the left image. Green, red and blue channels are eGFP-labeled cells expressing the helper viruses (green), rabies-virus expressing cells (red) and DAPI labeling (blue, only shown in the overlay images). Scale bars, 200 μm (*left*) and 50 μm (*right*). (**C, D**) Example

*Figure 7 continued on next page*

*Figure 7 continued*

widefield epifluorescence images of rabies virus-labeled presynaptic neurons in coronal sections, from a *Drd1a^Cre* mouse (**C**) and an *Adora2a^Cre* mouse (**D**). The abbreviations of brain areas are shown in *Figure 7—source data 2*. Scalebars, 250 μm. (**E, F**) Localizations of presynaptic neurons plotted in a 3D brain model, both for *Drd1a^Cre* (**E**) and *Adora2a^Cre* (**F**) mice. Each brain shows the results of two animals (blue and red dots) with one dataset being artificially mirrored on the other hemisphere for each genotype. For orientation, some brain areas are highlighted (S2 – green, InsCtx – yellow, PF – red, BLA – pink). (**G, H**) Quantification of labeled neurons for each brain area. Upwards and downward data signify cell density and absolute number of cells, quantified for N=2 mice of each genotype. Data from single animals are plotted as filled gray circles and triangles. (**I, J**) Distribution of presynaptic neurons along the anterior-posterior axis in the insular cortex (**I**), and in the secondary somatosensory cortex (**J**). The dashed line indicates the border between the anterior and posterior insular cortex according to *Franklin and Paxinos, 2016*.

The online version of this article includes the following source data and figure supplement(s) for figure 7:

**Source data 1.** Raw data for *Figure 7*.

**Source data 2.** List of abbreviations of brain areas.

**Figure supplement 1.** Control experiments for the specificity of helper-virus, and rabies virus expression.

**Figure supplement 2.** Example images for transsynaptically retrogradely labeled neurons in the BLA and PF.

**Figure supplement 3.** Analysis of the number of back-labeled neurons in *Adora2a^Cre* mice *relative* to *Drd1a^Cre*.

U=40, Mann-Whitney test; *Figure 8C–E*). At both connections, gradually increasing the stimulus light intensity led to a smooth increase of the EPSC amplitude, which shows that many axons from pInsCx neurons converge onto each type of principal neuron in the vTS (*Figure 8C and D*). At high light intensities, the optogenetically-evoked EPSC amplitudes saturated, suggesting that a maximal number of input axons was activated (*Figure 8C and D*). The paired-pulse ratio (PPR) did not differ between the two neuron types (*Figure 8F*; p=0.208, $t_{28}$=1.288, ; unpaired t-test; 95% CI: [0.637; 0.827] and [0.711; 0.960] for D1R+ and Adora+ neurons). On the other hand, the ratio of direct excitation over feedforward inhibition was significantly smaller in Adora+ vTS neurons as compared to D1R+ neurons (*Figure 8—figure supplement 1*; p=0.0054, U=31, Mann-Whitney test; 95% CI: [2.19; 4.10] and [1.05; 2.40] for D1R+ and Adora+ neurons). Taken together, optogenetically-assisted circuit mapping shows that both D1R+ and Adora+ neurons of the vTS receive robust excitatory inputs from the pInsCx. This, together with the monosynaptic rabies tracing (*Figure 7*), identifies the pInsCx as providing an important cortical inputs to the vTS.

## Fear learning causes opposite plasticity at cortical synapses on D1R+ and Adora+ neurons

The in-vivo imaging data showed that many D1R+ and Adora+ neurons in the vTS robustly respond to footshocks, and that during the course of fear learning, D1R+ neurons increase their responsiveness to the CS and to movement-ON transitions (*Figures 1–3*). Furthermore, silencing each vTS neuron population during the footshocks led to characteristic impairments of freezing in the presence, and absence of a learned CS (*Figures 5 and 6*). These findings suggest that plasticity takes place at synapses that drive D1R+ and Adora+ vTS neurons. We therefore next measured the AMPA/NMDA ratio and PPR following fear learning at the pInsCx-vTS D1R+/Adora+ synapses, to probe for postsynaptic or presynaptic forms of LTP induced by fear learning at each connection (*Yin et al., 2009*; *Shan et al., 2014*; *Rothwell et al., 2015*; *Lucas et al., 2016*; see *Palchaudhuri et al., 2022* for a review).

For this purpose, we used optogenetically-assisted circuit mapping to measure optogenetically evoked EPSCs at each connection. We injected *Drd1a^Cre* x *Rosa26^{LSL-tdTomato}* mice, and in a separate series of experiments *Adora2a^Cre* x *Rosa26^{LSL-tdTomato}* mice, with an AAV vector driving the expression of Chronos in neurons of the pInsCx. Three to 6 weeks later, the mice were subjected to auditory-cued fear learning (*Figure 9A*). Following the fear-memory recall session on day 3, which was performed to validate that mice had successfully learned the CS, mice were sacrificed, and optogenetically evoked EPSCs were recorded. A control group of mice underwent the same protocols, but no footshocks were applied ("CS only" group; *Figure 9A*). Optogenetically evoked EPSCs at the pInsCx to D1R+ vTS connection showed a significant increase of the AMPA/NMDA ratio in the CS+US group, as compared to the CS-only group, suggesting that at this connection, a postsynaptic form of LTP had occurred during fear learning (*Figure 9B and D*; p=0.001, $t_{21}$=3.833, unpaired t-test; 95% CI: [1.687; 2.764] and [3.128; 4.719] for CS-only and CS+US). The PPR, however, was unchanged in D1R+ neurons (*Figure 9C and D*; p=0.974, $t_{21}$=0.033, unpaired t-test; 95% CI: [0.448; 0.765] and [0.429; 0.791] for

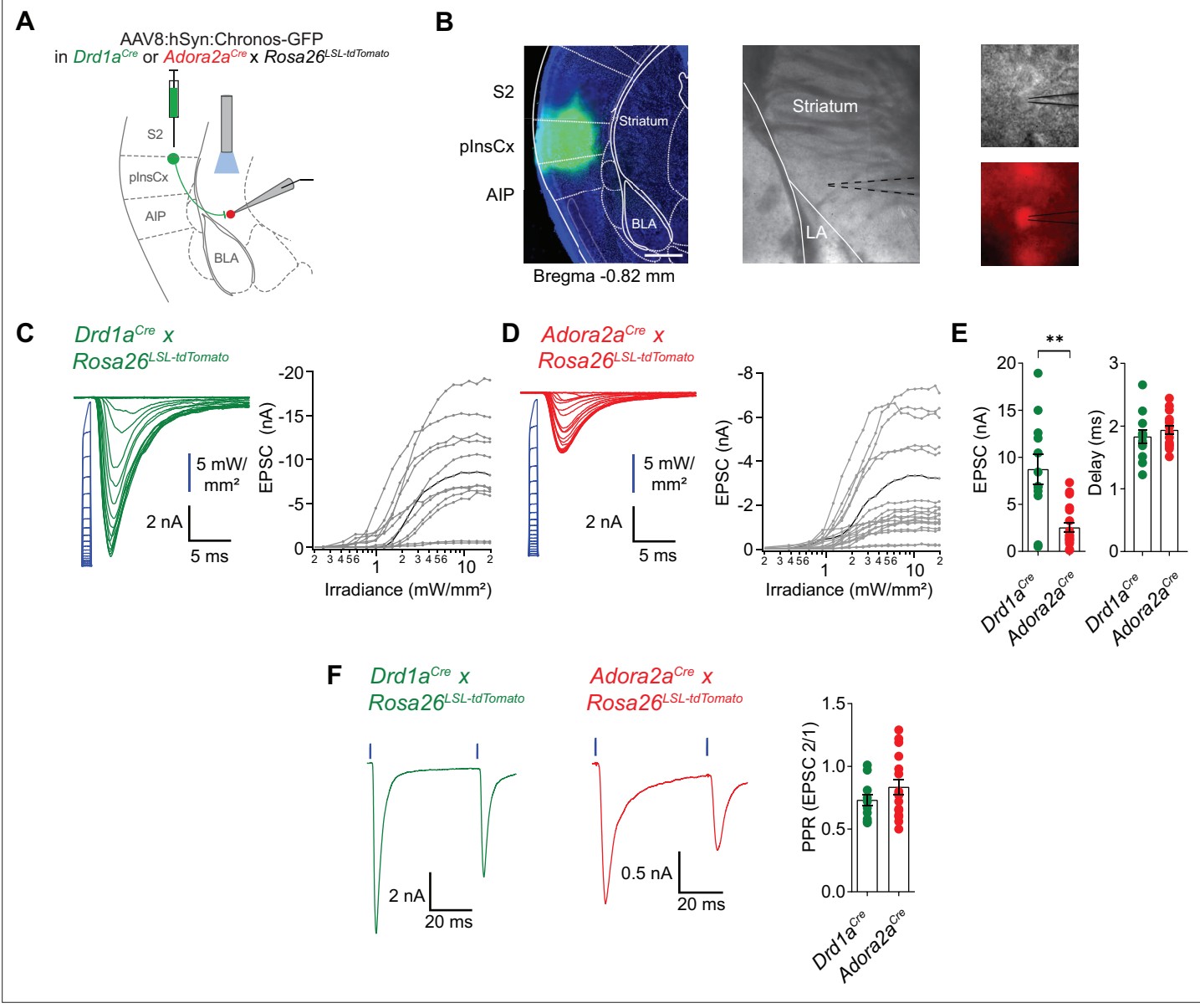

**Figure 8.** pInsCx provides strong excitatory inputs onto D1R+ - and Adora+ neurons in the vTS . (**A**) Experimental scheme of injection of an AAV vector driving the expression of Chronos into the pInsCx, and subsequent slice electrophysiology in the vTS. (**B**) *Left*, example fluorescence images of the injection site in the cortex expressing Chronos-eGFP (scalebar, 500 μm); *middle*, overview brightfield image with the position of the patch pipette (black dotted lines) in the vTS; *right*, higher magnification images of a recorded example cell (brightfield, *top*; and tdTomato fluorescence, *bottom*). (**C, D**) *Left*, EPSCs recorded by stimulating with 1ms light pulses of increasing intensities (blue, photodiode-recorded trace), and *right*, the resulting input-output curve of EPSC amplitude versus light intensity, with data from all recorded cells overlaid. The example cell shown on the left is highlighted in black in the right panel. Data is shown for D1R+ vTS neurons (C, n=12 recordings) and Adora+ vTS neurons (D, n=19 recordings). (**E**) Quantification of the maximal amplitude of EPSCs (*left*), and of the EPSC delay (*right*) measured in D1R+ - and Adora+ vTS neurons (n=12 and n=19 recordings, respectively). (**F**) Example traces (*left*, and *middle* panel) and quantification of PPR of optogenetically evoked EPSCs in D1R+ neurons (green; n=12 recordings) and Adora+ neurons (red; n=19 recordings).

The online version of this article includes the following source data and figure supplement(s) for figure 8:

**Source data 1.** Raw data and statistical tests for *Figure 8* and its supplements.

**Figure supplement 1.** Ratio of direct excitation to feedforward inhibition differs at the synapse from pInsCx to D1R+ vTS neurons versus Adora+ neurons.

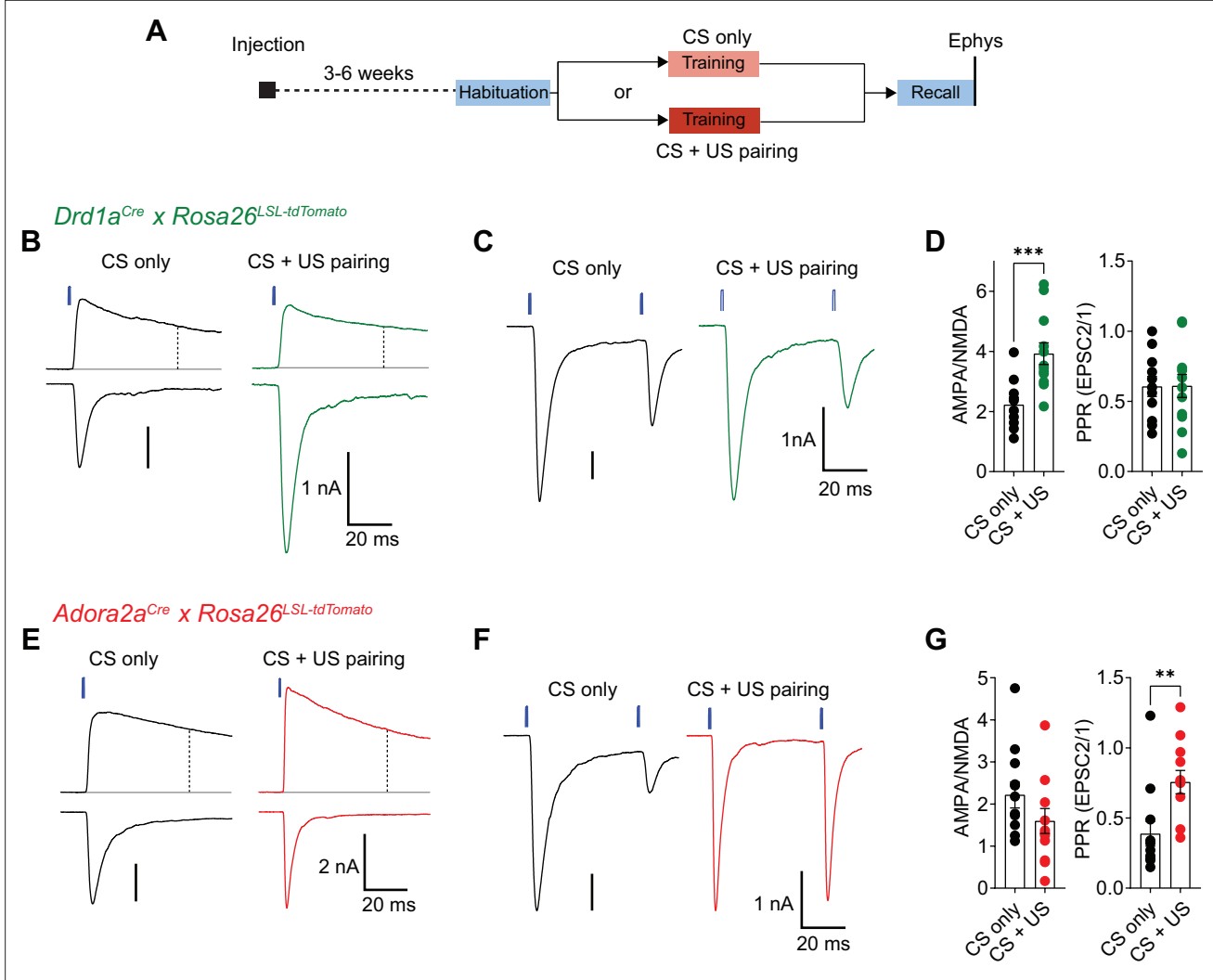

**Figure 9.** Fear learning induces long-term plasticity with opposite outcomes at pInsCx synapses onto D1R+ - and Adora+ vTS neurons. (**A**) Timeline of the ex vivo optogenetic assessment of plasticity following fear learning. See Results for details. (**B - D**) Measurements of AMPA/NMDA - ratios and PPR in two experimental groups ('CS-only' versus 'CS+US pairing') in *Drd1a^Cre* x *Rosa26^LSL-tdTomato* mice, to target the recording of direct pathway vTS neurons. (**B**) NMDA-EPSCs recorded at + 50 mV (top traces), and AMPA-EPSCs recorded at –70 mV (bottom traces), recorded in example D1R+ neurons from each experimental group. Vertical dashed line indicates the time of analysis of NMDA-EPSC. (**C**) AMPA-EPSCs (- 70 mV) recorded for the measurement of PPR (50ms inter-stimulus interval), shown for one example D1R+ neuron from each experimental group. (**D**) Quantification of AMPA/NMDA ratios and PPR recorded in each experimental group of *Drd1a^Cre* x *Rosa26^LSL-tdTomato* mice (CS only, n=11 recordings; CS+US pairing, n=12). Note the significantly increased AMPA/NMDA ratio in the CS+US pairing group as compared to the CS-only group (*P*=0.001; unpaired t-test), whereas the PPR was unchanged (*P*=0.97, unpaired t-test). (**E - G**) Measurements of AMPA/NMDA - ratios and PPR in two experimental groups ('CS-only' versus 'CS+US pairing') in *Adora2a^Cre* x *Rosa26^LSL-tdTomato* mice. (**E**) NMDA-EPSCs (top traces), and AMPA-EPSCs (bottom traces) recorded in example Adora+ neurons from each experimental group. (**F**) Example traces for the measurement of PPR of AMPA-EPSCs at - 70 mV, recorded in example Adora+ neurons of each experimental group. (**G**) Quantification of AMPA/NMDA ratios, and paired-pulse ratios recorded in each experimental group of the *Adora2a^Cre* x *Rosa26^LSL-tdTomato* mice (CS only, n=12 recordings; CS+US pairing, n=12). Note the significantly increased PPR in the CS+US pairing group as compared to the CS-only group (p=0.001; Mann-Whitney test). For further statistical parameters, see Results text.

The online version of this article includes the following source data for figure 9:

**Source data 1.** Raw data and statistical tests for **Figure 9**.

CS-only and CS+US). Conversely, at the pInsCx to Adora+ vTS connection, the AMPA/NMDA ratio was unchanged (*Figure 9E and G*; p=0.162, $t_{22}$=1.445, unpaired t-test; 95% CI: [1.538; 2.896] and [0.946; 2.251] for CS-only and CS+US), but instead, the PPR was increased in the CS+US group as compared to the CS-only group (*Figure 9F and G*; p=0.001, U=18, Mann-Whitney test; 95% CI: [0.193; 0.582] and [0.574; 0.938] for CS-only and CS+US). The latter finding suggests that fear learning induces a

presynaptic form of long-term depression at the pInsCx to Adora+ vTS synapse. Thus, auditory-cued fear learning drives long-term plasticity with opposite outcomes at cortical synapses onto D1R+ and Adora+ neurons in the vTS. These differential forms of long-term plasticity might contribute to the complementary roles of the two types of vTS principal neurons in fear learning.

## Discussion

Recent studies in mice have identified the posterior (tail) striatum as an anatomically, and connectionally separate part of the striatum, and evidence suggests that dopamine axons in the tail striatum code for salient sensory events (*Hintiryan et al., 2016*; *Hunnicutt et al., 2016*; *Menegas et al., 2018*; see *Valjent and Gangarossa, 2021* for a review). Here, we have investigated the role of the tail striatum in auditory-cued fear learning, in part motivated by earlier studies which showed a convergence of US - and CS signaling in the tail striatum located close to the LA (*Romanski et al., 1993*). We have targeted the ventral half of the tail striatum close to the amygdala, an area which we refer to as 'ventral tail striatum', vTS. This region might include the amygdala-striatum transition zone ('AStria'; located immediately adjacent to the amygdala), but our targeting was not limited to the AStria.

Using Cre-driver mouse lines to target D1R-expressing and Adora-expressing neurons of the direct and indirect pathways (*Gerfen et al., 2013*), we studied the role of each type of vTS principal neuron in auditory-cued fear learning. We found that both classes of vTS principal neurons were strongly activated by footshock stimuli, and coded for the movement state of the mice; smaller sub-populations of both neuron types also responded to tones in naive mice. Interestingly, the number of tone-responsive D1R+ neurons, but not of Adora+ neurons, increased during auditory-cued fear learning, not dissimilar to the acquisition of tone responsiveness of amygdalar neurons (*Quirk et al., 1995*; *Amano et al., 2011*; *Grewe et al., 2017*). To investigate the role of footshock-driven activity in vTS principal neuron, which most likely drives plasticity in these neurons, we optogenetically silenced neuronal activity at the time of footshock delivery. In *Drd1a^{Cre}* mice, this manipulation caused decreased auditory-cue driven freezing 1 day later, which leads us to conclude that footshock-driven plasticity in vTS D1R+ neurons contributes to the formation of a cue-driven fear memory. This conclusion is corroborated by our finding that fear learning induces a postsynaptic form of LTP at a major cortical input to D1R+ neurons in the vTS. On the other hand, silencing Adora+ neurons during the footshock led to a trend towards increased freezing in the absence of a learned CS, which suggests that indirect pathway neurons of the vTS have access to learned contextual freezing. Interestingly, the finding that the vTS Adora+ subsystem acts to suppress freezing in the *absence* of a danger-signaling CS, is reminiscent to aspects of safety learning (*Christianson et al., 2012*). Furthermore, a previous study has correlated the in-vivo activity of vTS neurons with safety learning, albeit without distinguishing between direct- and indirect pathway neurons (*Rogan et al., 2005*). Taken together, we find that direct and indirect pathway neurons of the vTS have different roles in fear learning. Both neuron types are expected to act synergistically to increase the difference in defensive behavior in the presence, and absence of a learned sensory cue.

In-vivo Ca$^{2+}$ imaging also revealed that sizable sub-populations of each type of vTS principal neurons code for movement-onset, and that in D1R+ neurons, these movement-ON responses were increased after fear learning (*Figures 1 and 3*). Movement-correlated activity has been observed in striatal neurons (*Markowitz et al., 2018*), as well as in several types of sensory cortices and in a more brain-wide fashion (*Niell and Stryker, 2010*; *Keller et al., 2012*; *Stringer et al., 2019*). At present, the role of the movement-related activity in the vTS remains unknown. Our optogenetic activation experiments did not reveal significant acute effects on the movement - or freezing behavior of mice, in line with previous findings in the posterior striatum (*Guo et al., 2018*), but different from results in the dorsomedial striatum (*Kravitz et al., 2010*) - the latter striatal area seems to have a more immediate role in movement control than the tail striatum. Therefore, it is unlikely that the movement-ON driven activity that we found here in D1R+ or Adora+ neurons of the vTS contributes to the control of movement. Rather, one could speculate that the movement-related activity in the vTS represents a signal related to the bodily state of the animal. In this regard, the strong excitatory input from the pInsCx to the vTS which we uncovered here seems relevant (*Figures 7 and 8*). Recent studies have shown that the pInsCx displays state-dependent signaling relating to physiological needs, or to the aversive state of the animal (*Livneh et al., 2017*; *Gehrlach et al., 2019*); it is possible that the activity

of the pInsCx is additionally modulated by movement-ON transitions of the animal. Thus, future work might investigate the origins, and functions of the movement-related signals in the vTS.

Previous studies in mice analyzed anterograde tracer experiments from cortex to striatum in a brain-wide fashion; this data has also revealed specific cortical input structures to the tail striatum (*Hintiryan et al., 2016*; *Hunnicutt et al., 2016*). We have used rabies-virus-mediated back-labeling from genetically identified neurons in the vTS, to identify presynaptic input neurons to the vTS more focally, and specifically for direct- and indirect pathway neurons (*Figure 7*; see also *Wall et al., 2013* for a similar approach in the more anterior dorsal striatum). Our results confirm previous studies regarding the innervation of the vTS by auditory cortical areas, and by amygdalar structures (*Hunnicutt et al., 2016*). Beyond this, our data identifies the InsCx and the S2 as major cortical input structures to the vTS, both for D1R+ and Adora+ neurons. We confirmed the functional relevance of the putative connections by optogenetically assisted circuit mapping, which revealed unusually large EPSCs at the input from the pInsCx to the vTS (note that the S2 was not covered in our ex vivo optogenetic experiments). Based on the graded input-output curves when varying the light intensity, these EPSCs likely reflect converging inputs from many pInsCx neurons (see also *Litvina and Chen, 2017*; *Gjoni et al., 2018*). The strong inputs from the pInsCx, and presumably from the S2, suggest that in addition to auditory- and visual information (*Yamamoto et al., 2012*; *Guo et al., 2018*; *Menegas et al., 2018*), somatosensory signals are processed in the vTS. For example, it is possible that the pInsCx transmits somatosensory or nociceptive information about a footshock during aversively motivated learning to the vTS; similarly, movement-related signals might be transferred at this connection, as discussed above. Moreover, it was shown that a subarea of the pInsCx processes auditory information (*Rodgers et al., 2008*; *Sawatari et al., 2011*; *Gogolla et al., 2014*), thus tone information might additionally be transmitted at this connection. Indeed, our finding of an enhanced AMPA/NMDA ratio following fear learning at the connection from the pInsCx to D1R+ vTS neurons gives rise to the hypothesis that learned auditory signals, or movement-related signals, are transmitted at this connection, a hypothesis which could be tested in future work.

Subtle, but functionally important differences in the brain areas that project to D1R+ versus Adora+ vTS neurons might determine their different in-vivo responses, and, in conjunction with the different output projections of direct- and indirect pathway neurons, their differential roles during fear learning. Recent work has shown that a separate set of midbrain dopamine neurons innervates the tail striatum, and that these dopamine projections are activated by salient, but not by rewarding stimuli (*Menegas et al., 2015*; *Menegas et al., 2018*). Thus, the differential roles of vTS D1R+ versus Adora+ neurons in fear learning are likely additionally shaped by the differential effects of dopamine on the two types of striatal principal neurons (*Gerfen et al., 1990*; *Tritsch and Sabatini, 2012*; *Hjorth et al., 2020*). Classical studies showed that dopamine release in the amygdala contributes to fear learning (*Lamont and Kokkinidis, 1998*; *Guarraci et al., 1999*; *Nader and LeDoux, 1999*), and a recent study showed that footshock-driven activity of ventral tegmental area (VTA) dopamine neurons that project to the basal amygdala (BA), contributes to fear learning (*Tang et al., 2020*). Therefore, dopamine signaling in the vTS, likely from a different source than dopamine release in the BA (*Menegas et al., 2015*; *Tang et al., 2020*), might contribute to diversifying the roles of vTS direct- and indirect pathway neurons in fear learning.

Fear learning shows sex-specific differences in rats (*Maren et al., 1994*; *Pryce et al., 1999*; *Gruene et al., 2015*; see *Lebron-Milad and Milad, 2012* for review). Thus, it was not possible to investigate female and male mice without differentiating between them; rather, we limited our study to male mice (see Materials and methods). In rodents, males in general show higher freezing than females, whereas females show a higher degree of fast bouts of escape reactions (called 'darting'), which can also take the form of learned responses (*Gruene et al., 2015*). Future work might investigate whether the direct- or indirect pathway neurons in the vTS contribute to sex-specific defensive behaviors like darting in females, and whether sex-specific differences exist in the circuit wiring and plasticity mechanisms of the vTS.

## Conclusions and outlook

In summary, we find that direct and indirect pathway neurons differentially modulate the degree of learned defensive behaviors in the presence, and absence of learned sensory cues. This uncovers a role of the vTS in balancing cue-specific reaction on the one hand, with a more generalized fear

response on the other hand. Adaptive fear discrimination is critically important for animal survival, but is dys-regulated in anxiety disorders (*Sangha et al., 2020*). Indeed, it was found that during fear generalization, PTSD patients exhibited stronger responses in the striatum, amygdala and insular cortex amongst other areas (*Morey et al., 2020*). Because fear expression in the absence of aversive sensory cues is downregulated by Adora+ neurons in the vTS, it is an intriguing possibility to harness the differential pharmacology of direct and indirect pathway neurons together with in vivo manipulations of plasticity (*Creed et al., 2015*), in an attempt to mitigate the effects of fear generalization.

# Materials and methods

## Key resources table

| Reagent type (species) or resource | Designation | Source or reference | Identifiers | Additional information |
|---|---|---|---|---|
| Strain, strain background (*M. musculus*) | C57BL/6J | The Jackson laboratory | Jax:000664 | |
| Genetic reagent (*M. musculus*) | Drd1a$^{Cre}$ | PMID:17855595; PMID:24360541 | MGI:4366803; MMRRC:030778-UCD; RRID:MMRRC_030778-UCD | STOCK Tg(Drd1-cre)EY217 Gsat/Mmucd |
| Genetic reagent (*M. musculus*) | Adora2a$^{Cre}$ | PMID:17855595; PMID:24360541 | MGI:4361654; MMRRC:036158-UCD; RRID:MMRRC_036158-UCD | B6.FVB(Cg)-Tg(Adora2a-cre)KG139Gsat/Mmucd |
| Genetic reagent (*M. musculus*) | Rosa26$^{LSL-tdTomato}$ | PMID:20023653 | Jax:007909; MGI:3809523; RRID:IMSR_JAX:007909 | B6.Cg-Gt(ROSA) 26Sor$^{tm9(CAG-tdTomato)Hze}$/J |
| Recombinant DNA reagent | AAV1:hSyn:FLEX: GCaMP6m (viral vector) | Viral vector facility, University of Zürich | ZurichVVF:v290-1 | (6.8e12/ml) |
| Recombinant DNA reagent | AAV1:hSyn:FLEX: Chronos-eGFP (viral vector) | University of North Carolina vector core | UNC:Boyden-AAV-Syn-Chronos-GFP | (2.8e12/ml) |
| Recombinant DNA reagent | AAV1:CAG:FLEX: Arch-eGFP (viral vector) | University of North Carolina vector core | UNC:Boyden-AAV-CAG-FLEX-Arch-GFP | (2.05e12/ml) |
| Recombinant DNA reagent | AAV1:CBA:FLEX: Arch-eGFP (viral vector) | University of Pennsylvania vector core | UPenn:AV-1-PV2432;Addgene:22222-AAV1 | (5.48e12/ml) |
| Recombinant DNA reagent | AAV1:CAG:FLEX: eGFP (viral vector) | University of North Carolina vector core | UNC:Boyden-AAV-CAG-FLEX-GFP | (4.4e12/ml) |
| Recombinant DNA reagent | AAV8:hSyn:Chronos-eGFP (viral vector) | University of North Carolina vector core | UNC:Boyden-AAV-Syn-Chronos-GFP | (6.5e12/ml) |
| Recombinant DNA reagent | AAV8:CAG:FLEX: tdTomato (viral vector) | University of North Carolina vector core | UNC:Boyden-AAV-CAG-FLEX-tdTomato | (6.5e12/ml) |
| Recombinant DNA reagent | AAV1:hSyn:FLEX:TVA-2a-eGFP-2a-oG (viral vector) | Viral vector facility, University of Zürich | ZurichVVF:v243-1; Addgene:85225 | (5.3e12/ml) |
| Recombinant DNA reagent | EnvA:deltaG–RV: dsRed (viral vector) | PMID:21867879 | Addgene:32638 | |
| Commercial assay or kit | Fluoroshield mounting medium with DAPI | Sigma Aldrich | Sigma:F6057-20ML | |
| Chemical compound, drug | gabazine | Abcam | Abcam:ab120042; Sigma:SR95531 | |
| Software, algorithm | VideoFreeze | Med Associates Inc | Med Associates:VideoFreeze | |
| Software, algorithm | EthoVision XT | Noldus Information Technologies | Noldus:EthoVisionXT13; RRID:SCR_000441 | version 13 |
| Software, algorithm | ezTrack | PMID:31882950 | RRID:SCR_021496 | https://github.com/denisecailab/ezTrack |
| Software, algorithm | Igor Pro | Wavemetrics Inc | RRID:SCR_000325 | version 7.08, 64 bit |
| Software, algorithm | Inscopix Data Processing Software | Inscopix Inc | Inscopix:IDPS | |
| Software, algorithm | CaImAn | PMID:30652683 | RRID:SCR_021152 | https://caiman.readthedocs.io/ |
| Software, algorithm | SHARP-Track | doi:10.1101/447995 | | https://github.com/cortex-lab/allenCCF |

*Continued on next page*

*Continued*

| Reagent type (species) or resource | Designation | Source or reference | Identifiers | Additional information |
|---|---|---|---|---|
| Software, algorithm | Brainrender | PMID:33739286 | RRID:SCR_022328 | https://edspace.american. edu/openbehavior/project/ brainrender/ |
| Software, algorithm | ABBA | doi:10.3389/fcomp. 2021.780026 | BIOP:ABBA | https://github.com/ BIOP/ijp-imagetoatlas |
| Software, algorithm | FIJI | PMID:22743772 | RRID:SCR_002285 | http://fiji.sc |
| Software, algorithm | Adobe Illustrator | Adobe Corporation | RRID:SCR_010279 | http://www.adobe.com/ products/illustrator.html |
| Software, algorithm | GraphPad Prism | GraphPad Software | RRID:SCR_002798 | version 9 |
| Software, algorithm | NeuroMatic | PMID:29670519 | RRID:SCR_004186 | plugin for IgorPro |
| Other | steretotaxic frame for small animals | David Kopf Instruments | David Kopf Instruments: Model 942 | used with Model 921 mouse adapter |
| Other | hydraulic one-axis manipulator | Narishige | MO-10 | for virus injections; see Materials and methods |
| Other | 600 µm / 7.3 mm ProView(TM) GRIN lens | Inscopix Inc | Inscopix:1050–004413 | used with nVista3.0 system |
| Other | nVista imaging system | Inscopix Inc | Inscopix:nVista3.0; RRID:SCR_017407 | for $Ca^{2+}$ imaging of neurons in freely moving mice; see Materials and methods |
| Other | optic fiber implants | Thorlabs Inc | Thorlabs:FT200EMT | 200 µm core / 0.39 NA / 230 µm outer diameter |
| Other | ceramic ferrule | Thorlabs Inc | Thorlabs:CFLC230 | 230 µm bore / 1.25 mm outer diameter |
| Other | blue light curing dental cement | Ivoclar Vivadent AG | Ivoclar Vivadent:Tetric EvoFlow | for securing implants at the skull surface |
| Other | light curing adhesive | Kulzer GmbH | Kulzer:iBond Total Etch | for treatment of skull before application of dental cement |
| Other | fear conditioning apparatus | Med Associates Inc | Med Associates:MED-VFC-OPTO-M | see Materials and methods |
| Other | electric footshock stimulator | Med Associates Inc | Med Associates:ENV-414S | used within the fear conditioning apparatus |
| Other | 561 nm solid-state laser | Changchun New Industries Optoelectronics Technology (CNI) | CNI:MGL-FN-561-AOM | fiber coupled, maximum output 100 mW; for in-vivo activation of Arch |
| Other | 473 nm solid-state laser | Changchun New Industries Optoelectronics Technology (CNI) | CNI:MBL-FN-473–150 mW | fiber coupled, maximum output 150 mW; for in-vivo activation of Chronos, see Materials and methods |
| Other | vibrating microtome VT1200S | Leica Microsystems | RRID:SCR_020243 | for preparation of brain slices; see Materials and methods |
| Other | patch-clamp amplifier EPC10/2 | HEKA Elektronik | RRID:SCR_018399 | for whole-cell patch-clamp recordings; see Materials and methods |
| Other | fluorescent microscope BX51WI | Olympus | RRID:SCR_018949 | to visualize neurons for whole-cell patch-clamp; see Materials and methods |
| Other | high-power LED, blue | Cree Inc | Cree:XPEBRY-L1-0000-00P02 | 460 nm; to excite Chronos in slices; see Materials and methods |
| Other | high-power LED, green | Cree Inc | Cree:XPEBGR-L1-0000-00D02 | 530 nm; to activate Arch in slices; see Materials and methods |
| Other | LED driver | Mightex Systems | Mightex Systems:BLS-1000–2 | |

*Continued on next page*

*Continued*

| Reagent type (species) or resource | Designation | Source or reference | Identifiers | Additional information |
|---|---|---|---|---|
| Other | silicone photodetector | Thorlabs Inc | Thorlabs:DET36A/M | to measure the time-course of LED light pulse in slice experiments; see **Figure 8C and D** and Materials and methods |
| Other | slide scanning fluorescent microscope | Olympus | Olympus:VS120-L100; RRID:SCR_018411 | for imaging post-hoc histology sections; see Materials and methods |
| Other | sliding microtome Microm HM450 | ThermoFisher Scientific | RRID:SCR_015959 | to prepare histological brain sections; see Materials and methods |
| Other | confocal microscope | Leica SP8 | RRID:SCR_018169 | for imaging post-hoc histology sections; see Materials and methods |

## Animals

The experiments were performed with different lines of genetically modified mice (*Mus musculus*) of male sex. The rationale for investigating exclusively male mice was as follows. The aims of the study were to investigate with optogenetic methods whether the vTS has a role in fear learning; to image the in-vivo responses of D1R+ and Adora+ neurons of the vTS during fear learning; to identify the main cortical inputs to both types of vTS principal neurons; and to study signs of long-term plasticity at cortical input synapses to both D1R+ and Adora+ neurons after fear learning. It has been shown, mainly using rats, that sex-specific differences exist in the strength and types of learned defensive behaviors (*Maren et al., 1994*; *Pryce et al., 1999*; *Gruene et al., 2015*). Therefore, including mice of both sexes in the study without differentiating between them would have most likely increased the variability of the results. Thus, it would have been necessary to include male *and* female mice in *separate* groups, which would have doubled the number of experimental groups, and experimental animals used in the study. We therefore decided to perform the initial study in male mice (see also Discussion).

The experiments were performed under authorizations for animal experimentation by the veterinary office of the Canton of Vaud, Switzerland (authorizations VD3274 and VD3518). The following mouse lines were used: (1) *Drd1a^Cre^* STOCK-Tg(Drd1-cre)EY217Gsat/Mmucd; see *Gong et al., 2007*; MMRRC: 030778-UCD; (2) *Adora2a^Cre^* B6.FVB(Cg)-Tg(Adora2a-cre)KG139Gsat/Mmucd; see *Gerfen et al., 2013*; MMRRC: 036158-UCD; (3) Cre-dependent tdTomato reporter line, *Rosa26^LSL-tdTomato^* (B6.Cg-Gt(ROSA)26Sor^tm9(CAG-tdTomato)Hze^/J; JAX stock #007909; also called 'Ai9'; *Madisen et al., 2010*). All mice strains were back-crossed for at least five generations to a C57BL/6J background. Mice were weaned at 21 days postnatally (P21), and groups of male mice were housed together under a 12/12 hr light/dark cycle (7:00 am, light on), with food and water ad libitum. Surgery was performed at P42 - P56; mice were separated into single cages one day before surgery. For behavioral experiments (*Figures 5 and 6*), mice from 1 to 2 litters were randomly assigned to control - (GFP-expressing) or effect group (Arch- expressing). Behavioral testing was performed during the light cycle.

## Viral vectors and injection coordinates

For in-vivo Ca²⁺-imaging experiments (*Figures 1–3*), we injected AAV1:hSyn:FLEX:GCaMP6m (200 nl; 6.8x10^12 vg[vector genomes]/ml; cat. v290-1; viral vector facility University of Zürich, Switzerland), into the left vTS of *Drd1a^Cre^* or *Adora2a^Cre^* mice, using the following coordinates: medio-lateral (ML) 3.2 mm; anterior-posterior (AP) –0.8 to –1.0 mm; dorso-ventral (DV) –4.3 mm (from bregma). For the optogenetic activation experiments (*Figure 4*), we injected AAV1:hSyn:FLEX:Chronos-eGFP (200 nl; 2.80x10^12 vg/ml; University of North Carolina - UNC vector core, Chapel Hill, NC, USA) bi-laterally into the vTS of *Drd1a^Cre^* or *Adora2a^Cre^* mice. For the optogenetic inhibition experiments (*Figures 5 and 6*), we injected AAV1:CAG:FLEX:Arch-eGFP (200 nl; 2.05x10^12 vg/ml; UNC vector core) bi-laterally into the vTS of *Drd1a^Cre^* or *Adora2a^Cre^* mice. In earlier experiments AAV1:CBA:FLEX:Arch-eGFP (200 nl;5.48x10^12 vg/ml; AV-1-PV2432; University of Pennsylvania vector core, Philadelphia, PA, USA, now at Addgene, 22222 - AAV1) was used; both Arch constructs correspond to the initially described 'Arch' (*Chow et al., 2010*). Mice in the control group received AAV1:CAG:FLEX:eGFP (200 nl; 4.4x10^12 vg/ml; UNC vector core).

For the ex-vivo optogenetically-assisted circuit-mapping experiments (*Figures 8 and 9*), *Drd1a*$^{Cre}$ x *Rosa26*$^{LSL-tdTomato}$ mice, or *Adora2a*$^{Cre}$ x *Rosa26*$^{LSL-tdTomato}$ mice were injected unilaterally into the pInsCx with AAV8:hSyn:Chronos-eGFP (200 nl; 6.5x10$^{12}$ vg/ml; UNC vector core), at the following stereotaxic coordinates: ML 4.2 mm; AP –0.55 mm, DV –3.8 mm (from bregma). In some experiments, *Drd1a*$^{Cre}$- or *Adora2a*$^{Cre}$ mice were used, and an AAV8:CAG:FLEX:tdTomato (200 nl; 6.5x10$^{12}$ vg/ml; UNC vector core) was additionally injected into the vTS (coordinates as above), to visualize *Drd1a*$^{Cre}$- or *Adora2a*$^{Cre}$-positive neurons for subsequent patch-clamp recordings.

## Surgery for virus injection, GRIN lens - , and optical fiber implantation

The surgery procedures for stereotactic injection of viral vectors alone, or combined with fiber implantation were as described in *Tang et al., 2020*. In short, a mouse was anesthetized with isoflurane (induction with 3%, maintained at 1%) and the head was fixed in a Model 940 stereotactic injection frame (David Kopf Instruments, Tujunga, CA, USA) using non-rupture ear bars (Zygoma Ear cups, Kopf Instruments Model 921). Local anesthesia was applied subcutaneously using a mix (50 µl) of lidocaine (1 mg/ml), bupivacaine (1.25 mg/ml) and epinephrine (0.625 µg/ml). After the skull was exposed, small craniotomies were drilled above the indicated coordinates for injection and fiber insertion. For fiber implantation, an additional craniotomy was made close to Lambda to insert an anchoring micro screw. Virus suspension was injected using pulled glass pipettes and an oil hydraulic micromanipulator (MO-10, Narishige, Tokyo, Japan) with an injection speed of ~60 nl/min. For in-vivo Ca$^{2+}$-imaging (*Figures 1–3*), a GRIN lens (600 µm / 7.3 mm ProView integrated lenses; cat. 1050–004413; Inscopix Inc, Palo Alto, CA, USA) was implanted 350 µm above the virus injection site. Prior to GRIN lens implantation, a 25 G medical injection needle was slowly inserted and retracted to facilitate the later insertion of the blunt-ended lens. To reduce deformation of brain tissue due to continuous vertical pressure, we lowered the GRIN lens with alternating down (150 µm) and up (50 µm) movements until the last 200 µm before the final position. For in-vivo optogenetic experiments (*Figures 4–6*), optical fiber implants were implanted bilaterally above the virus injection sites. The fiber implants were custom-made of a 200 µm core / 0.39 NA / 230 µm outer diameter optic fiber (FT200EMT; Thorlabs Inc, Newton, NJ, USA) secured inside 1.25 mm outer diameter ceramic ferrules (CFLC230; Thorlabs) as described in *Sparta et al., 2012*. The surface of skull, and the GRIN lens (or optical fiber implants) were treated with a light curing adhesive iBond Total Etch (Kulzer GmbH, Hanau, Germany). Blue light curing dental cement (Tetric EvoFlow, Ivoclar Vivadent, Schaan, Liechtenstein) was then applied to the skull to hold the GRIN lens (or optical fiber implants) in place. The open end of the GRIN lens at the integrated docking platform was sealed using a Kwik-Sil silicone compound (World Precision Instruments, Sarasota, FL, USA). After stitching, the skin was covered with Bepanthen Plus cream (Bayer AG, Leverkusen, Germany), the drinking water was supplemented with 1 mg/ml paracetamol and the animals were monitored for the following 6 days to ensure proper post-surgical recovery.

## Behavior

Auditory-cued fear learning was tested three to four weeks after the surgery. Mice were handled by the experimenter and habituated to the procedure of attaching a dummy miniature-microscope (Inscopix Inc) to the GRIN lens platform (in case of Ca$^{2+}$-imaging experiments), or optical patch cords (in case of optogenetic experiments), for 10–15 min on five consecutive days. An auditory-cued fear learning paradigm was performed in a conditioning chamber of a Video Fear Conditioning Optogenetics Package for Mouse (MED-VFC-OPTO-M, Med Associates Inc, Fairfax, VT, USA) under control of VideoFreeze software (Med Associates Inc). On day 1 (habituation), a mouse at a time was connected to the nVista3.0 mini-microscope (Inscopix Inc) or to the optic fiber patch cords, and the animal was placed in the conditioning chamber. The latter was a rectangular chamber with a metal grid floor, cleaned with 70% ethanol. During the ensuing habituation session, six tone blocks (CS), each consisting of 30 tone beeps (7 kHz, 80 dB, 100 ms duration, repeated at 1 Hz for 30 s), were applied 90 s apart. During a training session on day 2, the mouse was placed in the same chamber and presented with six CS blocks pseudo-randomly spaced 60–120 s apart, each followed by a 1 s foot shock US (0.6 mA, AC) delivered by a stimulator (ENV-414S, Med Associates Inc). During a fear memory recall session on day 3, the mouse was placed in a conditioning chamber within a different context, consisting of a curved wall and a smooth acrylic floor, cleaned with perfumed general-purpose soap, and four CS blocks were applied.

For optogenetic silencing experiments with Arch (*Figures 5 and 6*), light was delivered during 3 s starting 1 s before the footshock via 200 μm core / 0.22 NA optic fiber patch cords (Doric Lenses, Canada) from a 561 nm solid state laser (MGL-FN-561-AOM, 100 mW, Changchun New Industries Optoelectronics Technology, Changchun, China). The laser was equipped with an AOM and an additional mechanical shutter (SHB05T; Thorlabs). For optogenetic activation experiments with Chronos (*Figure 4*), 1 ms light pulses were delivered at 25 Hz, 2 s duration from a 473 nm solid-state laser (MBL-FN-473–150 mW, Changchun New Industries Optoelectronics Technology). The intensity of each laser was adjusted before the experiment to deliver 10 mW light power at the fiber tip.

The behavior of animals was recorded at video rate (30 Hz) by the VideoFreeze software (Med Associates Inc). Based on the behavioral videos, a movement trace was generated using ezTrack software (*Pennington et al., 2019*; see e.g. *Figure 1E–G*, red trace). The experimenter, and the person analyzing the data were blinded to the assignment of each mouse to the control - or test group. The movement index trace from ezTrack was used to compute a binary freezing trace using custom procedures in IgorPro 7 (WaveMetrics Inc, Lake Oswego, OR, USA). The animal was considered to be immobile (freezing state) if the movement index was below a threshold of 40 arbitrary units (without cable attachment) or 120 a.u. (with cable attachment), for a minimum duration of 0.5 s. The binned trace of percent time spent freezing (10 s bin size; see e.g. *Figure 5D and F*; *Figure 6D and F*) was calculated as a time-average of the freezing state from the binary trace, and then averaged across mice in each group.

## Microendoscopic Ca²⁺-imaging data acquisition and analysis

We used the nVista 3.0 system (Inscopix Inc) for imaging the activity of neurons expressing GCaMP6m in the vTS over the three-day fear conditioning paradigm. Fluorescent images were acquired at 30 Hz sequentially from three focal planes, resulting in an effective sampling rate of 10 Hz per plane. The intensity of the excitation LED in the nVista3.0 miniature microscope was set to 1–1.5 mW/mm², and the gain was adjusted to achieve pixel values within the dynamic range of the camera. The TTL pulses delivered from the behavioral setup (Med Associates) were digitized by the nVista 3.0 system to obtain synchronization between the mouse behavior and Ca²⁺-imaging data.

The initial processing of in-vivo Ca²⁺-imaging data was done using the Inscopix Data Processing Software (IDPS; Inscopix Inc). This included: (1) deinterleaving of the videos into the frames taken at individual focal planes; (2) spatial filtering; (3) motion correction; (4) export of the processed videos as TIFF image stacks; (5) export of the timestamps for each acquired frame, and for the experimental events such as CS and US timing. Next, the TIFF stacks were processed (except a movement correction step that was done by IDPS) using a Python-based package CalmAn (*Giovannucci et al., 2019*). The package is specifically optimized for the analysis of wide-field microendoscopic fluorescent Ca²⁺-imaging data using CNMF-E, an adaptation of constrained nonnegative matrix factorization algorithm (*Zhou et al., 2018*). Detection of ROIs for each focal plane was performed by CalmAn in an unsupervised manner, using the same set of analysis parameters across the three experimental days. This resulted in a set of background-corrected fluorescence traces for each automatically detected neuron, and also included a deconvolution step (*Pnevmatikakis et al., 2016*). The deconvolution returned the times, and amplitude values of Ca²⁺ events (see *Figure 1T*, gray vertical bars). The amplitudes of the deconvolved events were cumulated for each imaged neuron and each time epoch, to derive the amplitude-weighted frequency of Ca²⁺ events in one of four defined time epochs (during freezing, and in the absence of a CS [Frz_noCS]; during movement, and in the absence of a CS [Mov_noCS]; during freezing, and in the presence of a 30 s CS block [Frz_CS]; and during movement, and in the presence of a 30 s CS block [Mov_CS], e.g. see *Figure 1—figure supplement 5*).

Following the CalmAn analysis, the data were analyzed using custom routines in IgorPro 7 (WaveMetrics) as follows. Fluorescence intensity traces for each i-th cell ($F_i(t)$), were standardized by calculating Z-score traces as $Z_i(t) = \frac{F_i(t) - \langle F_i(t) \rangle}{\sigma(F_i(t))}$ , where $\langle F_i(t) \rangle$ and $\sigma(F_i(t))$ are the mean and standard deviation of the fluorescence intensity, respectively, calculated from the whole trace. Accordingly, the deconvolution traces were also normalized by the standard deviation of the respective fluorescent trace $\sigma(F_i(t))$ .

Prior to further analysis, any duplicate cells arising from different focal planes were identified with a semi-automated routine. In brief, the candidate duplicate cells were automatically short-listed based on the lateral proximity of their centers (<20 μm lateral distance) and high temporal cross-correlation

coefficient (>0.7) between their Z-score traces. Rejection of the duplicate cells featuring lower intensity signal (i.e. the cells more out of focus than the other) was validated manually.

The Z-score traces from retained cells were analyzed by their temporal alignments to the onsets of CS, US and movement-ON events as described in the Results (*Figures 1 and 3* and their supplements). Neurons were classified as 'responders' to a given event if the time-averaged Z-score value in the relevant time range after the event onset exceeded the chosen threshold. For US events, the range was 0–1 s and the Z-score threshold was 1.0. For the CS and movement-ON events, the range was 0–0.5 s and the threshold set to 0.2.

## Rabies tracing

For rabies tracing experiments (*Figure 7*) $Drd1a^{Cre}$ or $Adora2a^{Cre}$ mice were injected into the vTS (see above for coordinates) with a tricistronic vector; AAV1:hSyn:FLEX:TVA-2a-eGFP-2a-oG (250 nl; $5.3 \times 10^{12}$ vg/ml; cat. v243-1; viral vector facility University of Zürich) to render cells competent for EnvA-pseudotyped rabies virus uptake (*Wickersham et al., 2007*; *Wall et al., 2013*). In earlier experiments, we used a mix of AAVs (AAV8:hSyn:FLEX:TVA-2a-oG and AAV8:EF1α:FLEX:H2B-GFP-2a-oG; 250 nl; 1:1) for the same purpose. Three weeks later, the rabies vector EnvA:deltaG–RV-dsRed was injected at the same coordinates (250 nl; viral vector core Salk Institute for Biological Studies, La Jolla, CA, USA; *Osakada et al., 2011*). The animals were sacrificed 7 days later, and a histological analysis was performed on every second 40-µm-thick coronal section of the entire brain, from the level of the prefrontal cortex up to the end of the cerebellum. The resulting images were analysed in a semi-automatic fashion, that is dsRed-positive neurons were marked manually, brain sections were registered to the Allen brain atlas and marked neurons were automatically mapped onto the resulting brain regions using Matlab-based software (SHARP-Track; *Shamash et al., 2018*). The positioning of all long-range projecting cells was analysed and plotted on a 3D brain model using the python-based Brainrender software (*Claudi et al., 2020*).

## Electrophysiology

For whole-cell patch-clamp electrophysiology in slices, mice that had undergone surgery for AAV vector injection (see above) were sacrificed 3–6 weeks later. Mice were deeply anesthetized with isoflurane, and decapitated. The brain was quickly removed from the skull and placed in ice-cold preparation solution; the subsequent procedures followed the general method of *Ting et al., 2014*. The preparation solution contained (in mM): 110 N-methyl-D-glutamine, 2.5 KCl, 1.2 $NaH_2PO_4$, 20 HEPES, 25 Glucose, 5 Na-ascorbate, 2 Thiourea, 3 sodium pyruvate, 10 $MgCl_2$, 0.5 $CaCl_2$, saturated with carbogen gas ($O_2$ 95%/$CO_2$ 5%), pH of 7.4 adjusted with HCl. Coronal slices (300 µm) containing the vTS were cut using a Leica VT1200S slicer (Leica Microsystems, Wetzlar, Germany). Slices were stored for 7 min at 36 °C in the preparation solution and were then placed in a chamber containing a storage solution, composed of (in mM): 92 NaCl, 2.5 KCl, 30 $NaHCO_3$, 1.2 $NaH_2PO_4$, 20 HEPES, 25 glucose, 5 sodium ascorbate, 2 Thiourea, 3 Na-pyruvate, 2 $MgCl_2$ and 2 $CaCl_2$, pH 7.4 at room temperature, saturated with carbogen (*Ting et al., 2014*). Whole-cell patch-clamp recordings were performed with an extracellular solution containing (in mM): 125 NaCl, 2.5 KCl, 25 $NaHCO_3$, 1.2 $NaH_2PO_4$, 25 glucose, 0.4 Na-ascorbate, 3 Myo-Inositol, 2 Na-pyruvate, 1 $MgCl_2$ and 2 $CaCl_2$, pH 7.4, saturated with carbogen gas. The set-up was equipped with an EPC10/2 patch-clamp amplifier (HEKA Elektronik GmbH, Reutlingen, Germany), and an upright microscope (BX51WI; Olympus, Tokyo, Japan) with a 60 x / 0.9 NA water-immersion objective (LUMPlanFl, Olympus).

Patch-clamp experiments (*Figures 8 and 9*) were performed using a $Cs^+$-based intracellular solution (in mM): 140 $Cs^+$-gluconate, 10 HEPES, 8 TEA-Cl, 5 Na-phosphocreatine, 4 Mg-ATP, 0.3 Na-GTP, 5 EGTA, pH 7.2 adjusted with CsOH. AMPA/NMDA-ratio experiments were done in the presence of 5 µM $GABA_A$ receptor antagonist gabazine (SR-95531; Abcam, Cambridge, UK). Experiments to test the archaerhodopsin (Arch) properties (*Figure 5—figure supplement 2*) were performed using a $K^+$-based solution with (in mM): 8 KCl, 145 K-gluconate, 10 HEPES, 3 Na-phosphocreatine, 4 Mg-ATP, 0.3 Na-GTP, 5 EGTA, pH 7.2 adjusted with KOH. All electrophysiological experiments were conducted at near-physiological temperature (34 °C) using an inlet heater SHM-6, a heated recording chamber RC-26GL/PM-1 and a thermostatic control unit TC-344B (all from Warner Instruments, Holliston, MA, USA). All chemicals, unless indicated, were from Sigma-Aldrich (St. Louis, MO, USA).

For activation of the excitatory and inhibitory opsins in slice experiments, and for visualization of fluorophores in brain slices, high-power LEDs (CREE XP-E2, 460 nm and 530 nm; Cree Inc, Durham, NC, USA) were custom-coupled into the epifluorescence port of the microscope. Illumination was controlled by the EPC 10/2 amplifier DAC board connected to the LED driver (BLS-1000–2, Mightex Systems, Toronto, Canada). Irradiance was measured by a photodiode (DET36A/M, Thorlabs) coupled into the illumination light path, whose readings were calibrated by the light power measured under the 60 x objective using a power-meter model 1918-R equipped with a 818-UV detector (NewPort, Irvine, CA, USA). Electrophysiological recordings were analyzed in IgorPro (WaveMetrics) using the NeuroMatic plug-in (*Rothman and Silver, 2018*).

## Histology

For anatomical analysis of optic fiber- or GRIN lens positions, mice were transcardially perfused with a 4% paraformaldehyde (PFA) solution. The brains were post-fixed in PFA overnight and then transferred to 30% sucrose in phosphate-buffered solution for dehydration. Coronal brain sections of 40 μm thickness were prepared using a HM450 sliding microtome (Thermo Fisher Scientific, Waltham, MA, USA). Slices were mounted on Superfrost Plus slides (Thermo Fisher Scientific) and embedded in Fluoroshield mounting medium containing DAPI (Sigma-Aldrich) to stain cell nuclei. Slices were imaged with a slide scanning fluorescent microscope VS120-L100 (Olympus) with a 10 x /0.4 NA objective, or with a confocal microscope (Leica SP8). Brain atlas overlays are taken from *Franklin and Paxinos, 2016* and were fit to the brain section image using scaling and rotations in Adobe Illustrator (Adobe, San Jose, CA, USA). Where indicated, registration of brain section images was performed onto the Allen Brain Atlas using an open-source ABBA alignment tool for FIJI (https://github.com/BIOP/ijp-imagetoatlas), developed at the Bioimaging and Optics Platform (BIOP) at EPFL (*Chiaruttini et al., 2022*). The majority of brain structure names and their abbreviations follows *Franklin and Paxinos, 2016*; their correspondence to the Allen brain atlas are given in *Figure 7—source data 2*.

## Statistical analysis

Statistical analysis was performed in GraphPad Prism 9 (GraphPad, San Diego, CA, USA). Before choosing the main statistical test, the distribution of the data was tested for normality using a Shapiro-Wilk test. When normality was confirmed, we used a paired or unpaired version of the two-tailed Student's t-tests for two-sample datasets, as indicated. For the comparison of relative datasets, we used a one-sample two-tailed t-test. When the data was not normally distributed, we used two-tailed non-parametric tests: a Wilcoxon matched-pairs signed-rank test for paired comparisons, or Mann-Whitney U test for unpaired comparisons of two-sample datasets, as indicated.

For datasets with more than two samples (in-vivo Ca$^{2+}$ imaging data; *Figures 1 and 3*), the data showed skewed distributions which did not pass the normality test, therefore we used a non-parametric version of a one-way ANOVA, a Kruskal-Wallis test. If this test detected significant differences, it was followed by Dunn's post-hoc test for multiple comparisons (called Dunn's MC test). For datasets influenced by two factors such as the optogenetic silencing/control group and the time of the experiment (*Figures 5 and 6*), we used a repeated-measures two-way ANOVA (RM-ANOVA) separately for the training and fear retrieval days. If RM-ANOVA reported significance, it was followed by Šidák's post-hoc tests for the respective factor.

The change in number of neurons responding in-vivo to different events across training (*Figure 1L, S*, *Figure 3I, P*) was statistically assessed using a Chi-square test. For the optogenetic silencing experiments, we determined the sample size a priori using a G*Power software (*Faul et al., 2007*) for an RM-ANOVA test (two groups, 4 replicate measurements for the fear retrieval day) assuming an average change in freezing of 20% (from 60 to 40%) and a standard deviation of 15% (resulting in an effect size of 0.667), significance level $\alpha$=0.05, power 1-β=0.85, and the correlation between repeated measures of 0.5. The resulting total sample size was N=16 (N=8 mice per group), with a critical $F_{crit}$ = 4.6. There was no sample size estimation made for other experiments.

Each specific statistical test is mentioned in the Results text, and the input data along with statistical test summary (such as p-values, values of test statistics, degrees of freedom, etc.) and the main descriptive statistics are given in the statistics Tables for each relevant figure. The data are expressed as mean ± SEM. Statistical significance, if applicable, is indicated in the Figures using asterisks as p≤0.05 (*), p≤0.01 (**) and p≤0.001 (***).

## Acknowledgements

We thank Shriya Palchaudhuri and Denys Osypenko for helpful discussions regarding the analysis of freezing behavior, and Elena Mombelli for help with anatomical experiments. Images were acquired at the Bioimaging and Optics Platform of EPFL (BIOP). This work was supported by an EMBO fellowship to MK (# ALTF 224-2015), and by grants from the Swiss National Science foundation (SNSF; grant number 31003 A_176332/1) and from the National Competence Center for Research (NCCR) of the SNSF "Synapsy - The Synaptic Bases of Mental disease", project P28 (both to RS).

## Additional information

### Funding

| Funder | Grant reference number | Author |
| --- | --- | --- |
| European Molecular Biology Organization | ALTF 224-2015 | Michael Kintscher |
| Swiss National Science Foundation | 31003A_176332 / 1 | Ralf Schneggenburger |
| NCCR Synapsy - The Synaptic Bases of Mental Disease | Project P28 | Ralf Schneggenburger |

The funders had no role in study design, data collection and interpretation, or the decision to submit the work for publication.

### Author contributions

Michael Kintscher, Conceptualization, Formal analysis, Funding acquisition, Validation, Investigation, Visualization, Methodology, Writing – original draft, Writing – review and editing; Olexiy Kochubey, Formal analysis, Validation, Investigation, Methodology, Writing – review and editing; Ralf Schneggenburger, Conceptualization, Supervision, Funding acquisition, Validation, Writing – original draft, Writing – review and editing

### Author ORCIDs

Michael Kintscher http://orcid.org/0000-0003-2355-1369
Olexiy Kochubey http://orcid.org/0000-0002-8115-7733
Ralf Schneggenburger http://orcid.org/0000-0002-6223-2830

### Ethics

All experimental procedures with laboratory animals (Mus musculus) were performed under authorizations for animal experimentation by the veterinary office of the Canton of Vaud, Switzerland (authorizations VD3274 and VD3518).

### Decision letter and Author response

Decision letter https://doi.org/10.7554/eLife.75703.sa1
Author response https://doi.org/10.7554/eLife.75703.sa2

## Additional files

### Supplementary files
• Transparent reporting form

### Data availability

The underlying raw data leading to the conclusions of this paper is available at Zenodo data repository https://doi.org/10.5281/zenodo.7530512.

The following dataset was generated:

| Author(s) | Year | Dataset title | Dataset URL | Database and Identifier |
|---|---|---|---|---|
| Kintscher M, Kochubey O, Schneggenburger R | 2023 | A striatal circuit balances learned fear in the presence and absence of sensory cues | https://doi.org/10.5281/zenodo.7530512 | Zenodo, 10.5281/zenodo.7530512 |

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
