## [Editor Report]

This important study examines the contribution of an understudied brain region to fear conditioning in mice. The evidence supporting the authors' conclusions is convincing. This paper will interest neuroscientists working in the fields of basal ganglia, amygdala, and fear learning.

---

## [Decision Letter]

**Decision letter after peer review:**

Thank you for submitting your article "A striatal circuit balances learned fear in the presence and absence of sensory cues" for consideration by *eLife*. Your article has been reviewed by 3 peer reviewers, and the evaluation has been overseen by a Reviewing Editor and Kate Wassum as the Senior Editor. The reviewers have opted to remain anonymous.

Essential revisions:

The individual assessments and recommendations from each of the reviewers are included below. In addition, here we provide you with a list of items that we collectively consider to be essential revisions that must be addressed in order for the manuscript to be considered further for publication in *eLife*. Please also address the individual points raised by each of the reviewers in the public reviews and recommendations for authors, as we consider that addressing them will strengthen the overall quality of the paper. Our requested Essential Revisions are:

1) Revise the overall presentation of the manuscript to avoid the use of inaccurate wording and/or conclusions not supported by the findings presented on the manuscript. Please refer to the individual assessments from the reviewers for specific guidance on this.

2) Appropriately frame the considerations when defining the boundaries of the AStria and how this area is distinguished from the neighboring tail of the striatum. A more comprehensive discussion of anatomical features and connectivity of the AStria is also required. Please refer to comments from Reviewer 1. Maps of expression spread for each subject (e.g., overlaid) and lens or fiber placement would also help provide transparency for readers in the location of the recordings and manipulations.

3) Reanalyze the imaging data in accordance to the comments from Reviewers 2 and 3.

4) Address the reviewers' concerns related to the AP deconvolution method employed by the authors.

5) Include important missing controls as highlighted by the reviewers (e.g., positive control for optogenetic stimulation experiments, negative control for the rabies tracing data).

6) Provide a rationale for why only male subjects were included in the study. Specifically, the authors should: a) state the sex in the abstract, b) explain why only one sex was used in the methods, and c) acknowledge and discuss the limitation of the exclusion of one sex in the discussion.

7) Demonstrate that the Cre lines used are valid as tools to achieve genetic access to the neuronal populations of the AStria.

8) Please ensure your manuscript complies with the *eLife* policies for statistical reporting: https://reviewer.elifesciences.org/author-guide/full "Report exact p-values wherever possible alongside the summary statistics and 95% confidence intervals. These should be reported for all key questions and not only when the p-value is less than 0.05." This should be reported in the main text.

9) Please include a key resource table in your methods and clarify whether animals were tested in the light or dark cycle.

*Reviewer #1 (Recommendations for the authors):*

Thalamic (PIN/PIL) and cortical regions (TeA/Auv) are missing or only weakly present when the connectivity of Astr is described. Romanski and Ledoux (1993) showed strong TeA-Astr connectivity, and Ledoux et al. (1990) as well as Barsy et al., (2020) highlighted a strong thalamo-Astr projection from the area of PIL and SG. Furthermore, the latter study also investigated the CS, US and CS/US responsiveness of Astr neurons which were found to be (at least, partially) mediated by the thalamic inputs.

The LA-Astr projection seems to be also underestimated in the view of the earlier publications (eg. Jolkkonen et al., 2001).

One of the reasons causing these connectivity differences could arise from the targeting of Astr. Identification of Astr territory is not trivial, since no marker is available which could differentiate AStr from the rest of the caudal (tail) striatum. Still, as earlier studies showed that these two striatal regions may receive distinct inputs (eg. MGN to the tail, while PIN/PIL to the Astr; LA mostly targets Astr and not the other parts of the caudal striatum), it is important to investigate the same and consistent striatal region. Paxinos mouse brain atlas indicates AStr from bregma ~-0.9 (AP) between LA and LGP (and more caudal, the stria terminalis) and right above the CeA. The dorsal border is uncertain, but mostly indicated ventral to the top 'corner' of the LA (like in a recent review by Valjent and Gangarossa, 2020). However, many representative images and schematic drawing highlight more dorsal and anterior regions to be targeted which resemble the ventral part of the striatal tail, rather than Astr. While most of the foot shock and CS responsive cells were located in the posterior part of the examined regions (more posterior from bregma -1; Figure 1-3), some injection sites (AAV, RV) and fiber optic locations were more anterior. Indeed, the authors also use the tail of the striatum term. As no comprehensive functional study is present (to my best knowledge) which simultaneously investigate the tail of the striatum and Astr, it could cause many discrepancies if the two regions are mixed.

Thus, some of the discrepancies could have arisen from targeting the tail of the striatum rather than Astr.

Several recent (Barsy et al., 2020, Gilad et al., 2020; Talyor et al., 2021) and earlier studies (Bordi and Ledoux et al., 1994a,b; Apergis-Schoute et al., 2005; Han et al., 2008; Weinberger, 2011, etc) suggested that CS-US association and thus, neuronal plasticity can occur prior to the amygdala, in non-sensory (higher-order) thalamic regions, like PIN/PIL. Therefore, the following statement is only partially correct. "In the lateral (LA)- and basal amygdala (LA and BA respectively), auditory (CS) and somatosensory (US) information are integrated.."

Was there any level of dorsoventral topography found in the location of the Cs and/or US responsive cells (striatal tail vs. Astr)?

Is it possible that the examined and responded D1 and Adora+ cells are actually located in two distinct striatal regions? It is discussed that these are rather topographically non-overlapping populations.

Is it possible to follow the Ca-activity of individual cells throughout the entire learning paradigm? If yes, plasticity of a single cell could be analysed (like in the thalamus, Taylor et al., 2021).

As Figure 7 showed some level of topography in the InsCx-Astr connectivity, were different sectors of InsCx targeted in the D1 and Adore-Cre animals used for patch clamp recordings in Figure 8?

Pg 17. Between the AAV and RV, three weeks is indicated in the text, but 4 weeks in the figure.

Figure 8, Figure suppl. 4 B and C. It seems that distinct parts of striatum are targeted.

As there is very little data about Astr, it would be interesting to examine the basic anatomical organization of Astr. For example, the proportion of D1+ and Adora+ cells, the presence of other types of neurons (cholinergic, PV cells, etc). But, these may be far beyond the scope of the present study.

*Reviewer #2 (Recommendations for the authors):*

It is disappointing that the presentation and some of the analyses do not do justice to the work. Please see my points below. In particular, the heavy handed analyses of the fluorescence transients is too much and fraught with potential for error. More importantly, it obscures what appear to be large differences that could be extracted more easily (and convincingly). Please see the "approach section below". Lastly, at times the authors interpret their data and results too optimistically.

Specifically, the authors should:

1. Reanalyze the data using only z-score F(t).

2. Show more primary data – images of cells, etc…

3. Refrain from reaching conclusions too early in the results especially that it is the contribution of plasticity of the neurons that is being studied in the silencing experiments (as opposed to the need for activity).

4. Improve the presentation of the methods

Please read the detailed comments below.

Abstract:

1. Imprecise wording. The notion of "balancing" is unclear in "defensive behaviors need to be finely balance" which makes it hard to understand exactly what is being studied.

2. "In-vivo optogenetic silencing during the training day showed that plasticity in D1R+ AStria neurons contributes to auditory-cued fear memories"

Please see my comments on the interpretation of the silencing experiments.

Results:

1. Show some images of neurons with corresponding fluorescence transients to understand the quality of the data. I can't find them anywhere in the figures.

2. Line 111 – "Seemed to increase". What does this mean?

3. Line 114 – Claim to have derived APs by deconvolution.

4. Line 127 – "Similarly suggested an increased".

5. See comments on methods – by the time we get to summary data as I Figures IJ-L it is very hard to understand what these mean. They are in au units x 10e-4. What does this mean? If they are au then they can be remapped.

6. Line 154 "These experiments thus show that D1R+ AStria neurons increase their AP firing activity during the onset of movements after a period of immobility. "

The analysis in this section is potentially problematic as tones are being played and the animal is starting and stopping movement. It is not clear that the response is to the movement change as opposed to the tone. Please show the histogram of tone start and stop times relative to the movement onsets.

A better analysis is one that models the contributions tones, movement starts, stops, and US. This can be done with a GLM and will alone one to disambiguate these confounds.

I am especially worried because of the follow-up sentence "Line 155 were more pronounced during the CS than in the absence of a tone" which suggests that these are not movement cells.

7. Line 159 – The conclusion in this sentence cannot be justified based on the data shown so far.

8. Line 168 – One cannot make these conclusions by comparison of event-triggered averaging.

9. Overall Figure 1 is very hard to follow. The number of panels should be reduced, labels should be placed to show which analyses are relative to movement, CS, etc…

10. Figure 2 is very nice.

11. The paragraph starting on line 252 is confusing. Are the authors saying that the # of cells responding goes up (i.e. response fidelity) but that the response per active cell (estimated AP content) does not? Both the positive and negative conclusion refer to Figure 3R. I can't find a definition of response fidelity, which is used several times.

12. The analysis of Figure 3S says that significance was judged by a KS-test, which is usually used for comparisons of unbinned cdfs. The graph is an average +/- error bar. How was the KS test used?

13. If the freezing habituation data in 1S and 3S cannot be analyzed statistically, it should be removed from the panels as one cannot help but interpret the data shown quantitatively.

14. The Venn diagrams in 3V and 2D would benefit by labeling the numbers in the main intersection areas.

15. In figure 4, the baseline movement of the Adora2a and D1R groups is very different. Are the two genotypes equivalent? These are BACs so it would not be too surprising.

16. For Figure 4, what in vivo proof is there that the cells are being activated? Given the negative conclusion, such a control is important.

17. The experiments motivated by the paragraph starting at Line 325 are very nice. However, the paragraph is troubling. The manipulation is to reduce activity and it should be phrased in that way. Instead, a hypothesis based on plasticity is presented, from which the experimental manipulation of activity is indirectly motivated.

18. There are differences in baseline freezing (5F) before the light is turned on that are of similar magnitude (relative to the low level of freezing) as seen in 5H. Similarly, the comparison between no effect in 5G right to less effect in 5I left may not be fair given that the latter is in "steady state" and the former not. Is an RM-ANOVA the right way to go?

19. Similar concerns exist for the data in Figure 6.

Can the differences in Figure 5 or 6 be used to identify individual mice as ones that received silencing and ones that did not? It seems unlikely, especially for Figure H. It might have been better to run cohorts of mice in which the silencing was only applied in the retrieval portion. Otherwise, it is unclear if the relatively modest effects are due to a difference during training that is not reflected in freezing rates.

20. The main text does not reference controls for the rabies experiments which generally should include (1) TVA dependence of rabies infection, (2) G dependence of spread, (3) Cre-dependence of the DIO/floxed constructs. It is important to do these with the same batch of viruses used for the experiments (ideally in parallel!).

21. I hesitate to make too much of the differences shown in 7I/J given N=2, especially without a careful analysis of all the starter cells to show that the number and distribution of these were the same across genotypes.

22. The currents in Figures 8G-H are huge and loss of voltage clamp must have occurred, especially when measuring NMDA receptor currents.

Approach:

1. Large lesions even for optogenetics because of use of 1.25 mm cannula.

2. Line 763 -- "If mice from any experimental group showed low freezing, they were excluded from the analysis; we used a threshold of 20% time spent freezing

during the fifth and sixth CS-blocks on day 2 (7 out of 59 mice for the behavioral experiments were excluded)."

It is good of the authors to give the exclusion criteria. However, this is worrisome. Some of the loss of function experiments are based on the hypothesis that activity is necessary for learning. Therefore, this discards the poor learners, right?

What is the breakdown of exclusion across the different experimental and genetic conditions? Why not simply include all animals to capture the natural variability?

3. The deconvolution approach is worrisome for a few reasons.

a. Ca entry into these neurons is highly non-linear with single AP, burst AP, up-state APs, and GPRC-mediated influx. It is unclear that one can use a linear model of # of APs to [Ca] (to F(Gcamp6m)). It does not seem that this step is necessary (especially given the heavy filtering).

b. Data is acquired at 10Hz but then a band-pass filter at 2-4 Hz (If I read it correctly although the methods are not very clear). This is much less than the Nyquist limit. Why? Also why place a low-frequency limit (i.e. why use a band pass as opposed to a low-pass to simply get rid of shot and electronic noise?).

c. The deconvolution kernel is arbitrary and uses a rise time that cannot be captured with the 2-4 Hz band pass. The authors need to show that 0.5 s tau(decay) is justified. There is no justification for choosing an amplitude of 2.

d. Line 793 – The phrase "deconvolution yielded a… proportional to AP firing rate" must be removed as it cannot be justified.

e. The deconvolved signal (R) is further filtered with a box car – Why? The filtering is done upstream on the fluorescence. It should not be done again.

f. "local peaks" exceeding a threshold (arbitrarily set to 2) were detected using a "first-derivative" method. Why add a threshold here? It seems unjustified. Second, how does one detect a peak with a first-derivative? Shouldn't it be a second derivative? Or are they detecting fast-rate of rise, in which case why set a peak amplitude threshold?

4. All of these concerns are acknowledged later (line 801) in "we did not aim to infer exact AP spiking rates of cells". Then why do this and present it in this way?

5. They should simply get rid of all these steps and analyze the z-score raw F (calculating a DF/F and then z-scoring is no different than just z-scored the Z).

6. y comparing the

7. The section starting on line 809 indicates that different baselines were chosen for comparison for different types of cells. This introduces a circular bias. One needs to be able to statistically show that a cell is a "move-on" or CS responder without changing a baseline. Why not use the Z-score values in the time-bin without picking a base-line – i.e. compare activity in that window to all activity?

8. Line 798 – The amplitudes of the events from deconvolution (referred to as estimated AP content, EAC [a.u./s]) were proportional to the amplitude of the ca^2+^-transients (e.g. see Figure 1 —figure supplement 1E).

a. I can't find the referenced data in a figure. The sentence suggests a scatter plot with some kind of regression analysis?

b. If this is true, why do it all? Just use the F as suggested above and save all these potentially problematic analysis steps.

*Reviewer #3 (Recommendations for the authors):*

1. The study used only male mice. There is no basis for excluding females.

2. The authors did not verify specificity of D1R and Adora lines (tdTomato + RNAscope for D1R and Adora). Without this verification, it is unclear if the fluorescent patterns reported actually reflect patterns of D1R and Adora neurons.

3. Inspection of the heat plots shown in figures 1 and 3 indicate the fluorescent data contains repeating 'neurons'. For example in Figure 3, the two signals in rows 10 and 11 are identical. This is particularly notable because the repeating 'neuron' shows the largest change in fluorescence. In addition, rows ~17 and ~34 appear virtually identical as well. This is a major problem and indicates the authors have an issue in their analysis pipeline. Greater care needs to be taken to ensure repeating neuron's are not reported and analyzed.

Same issue occurs in Figure 1. Rows 3 and 4 are identical signals. Repeating 'neurons' appears to be a problem throughout.

4. GCaMP6m is a fluorescent calcium indicator. While calcium entry into neurons is required for AP-driven vesicular release, calcium entry into a neuron can reflect many other processes. In this manuscript, the authors wish to examine single unit activity by acquiring and deconvolving fluorescent signals. In fact, there are known issues relating fluorescent signals to action potentials in the striatum:

https://www.biorxiv.org/content/10.1101/2021.01.20.427525v2.full

If the authors wish to analyze spiking activity of the striatal neurons, they need to use techniques that directly record action potentials. These techniques are readily available. If the authors wish to record fluorescent changes resulting from calcium influx, they need to embrace this decision and only analyze changes in fluorescence data.

5. I found it very difficult to track the logical flow of the analyses. The authors start off by showing 'neurons' are responsive to cues. But then almost immediately pivot to showing they do not in fact respond to cues, but movement during cues. A lot of figure space and analysis is devoted to this, such that the big picture is lost. I would find it more convincing to develop analyses that directly compare cue vs. movement vs. cue x movement responding from the outset. A regression approach may be useful. Take the 30 s prior to cue and 30 s cue. Have one regressor be cue on vs cue off, another be movement on vs movement off and the last be the intersection of cue on/off and movement on/off. This would simultaneously compare each regressor to each 'neurons' activity pattern, determining which best captures change in fluorescence.

6. Do not show hypothetical behavior data (as in Figure 1A). Only show real behavior data.

In its present form, I do not feel that the authors achieved their aims or that the results support their conclusions. Addressing the points above is more likely to produce such results. Although even then, it appears that the contribution of these two cell types to tone-shock learning appears limited. If these neurons are contributing to aversive behavior, perhaps their contribution would be better captured by procedures in which movement is a more central element to behavior.

[Editors' note: further revisions were suggested prior to acceptance, as described below.]

Thank you for resubmitting your work entitled "A striatal circuit balances learned fear in the presence and absence of sensory cues" for further consideration by *eLife*. Your revised article has been evaluated by Kate Wassum (Senior Editor) and a Reviewing Editor.

The manuscript has been greatly improved during the revision, but there are some issues that remain to be addressed prior to acceptance, as outlined below:

*Reviewer #2 (Recommendations for the authors):*

The authors have greatly altered and improved the manuscript to address the reviewer concerns. In particular, they have implemented a standardized Ca analysis pipeline, altered the description of the conclusions in many places, improved the presentation of the data and added some necessary controls.

I wish that they had done a full characterization of the RV controls, instead of showing some selected images, but that is ok.

I am confused about the sentence (lines 199-201)

"These experiments show that a substantial sub-population of Adora+ neurons in the vTS codes for movement onset, but this representation was unchanged by fear learning (Figure 3M – P)."

The changes in event frequency in Figure 3R from habituation to training for freeze state are quite large yet the conclusion is that training does not influence the representation of movement in Adora2a+ neurons? Perhaps I am misinterpreting the statement.

---

## [Author Response]

Essential revisions:The individual assessments and recommendations from each of the reviewers are included below. In addition, here we provide you with a list of items that we collectively consider to be essential revisions that must be addressed in order for the manuscript to be considered further for publication in eLife. Please also address the individual points raised by each of the reviewers in the public reviews and recommendations for authors, as we consider that addressing them will strengthen the overall quality of the paper. Our requested Essential Revisions are:1) Revise the overall presentation of the manuscript to avoid the use of inaccurate wording and/or conclusions not supported by the findings presented on the manuscript. Please refer to the individual assessments from the reviewers for specific guidance on this.

The manuscript has been re-written at many points, to address each point of criticism by the reviewers (see our answers to the individual points raised by each reviewer). We hope that the revision has improved the paper.

2) Appropriately frame the considerations when defining the boundaries of the AStria and how this area is distinguished from the neighboring tail of the striatum. A more comprehensive discussion of anatomical features and connectivity of the AStria is also required. Please refer to comments from Reviewer 1. Maps of expression spread for each subject (e.g., overlaid) and lens or fiber placement would also help provide transparency for readers in the location of the recordings and manipulations.

We agree with the point made by reviewer 1, point [3] about the difficulty in demarcating the AStria / ventral tail striatum (vTS). ("One of the reasons causing these connectivity differences could arise from the targeting of Astr. …"). Because of the additional concern whether the Cre mouse lines used here are well-suited to drive Cre-expression in the AStria / vTS ("Essential revision point #7, below, and reviewer 3, point 2), we performed further anatomical experiments. For this, we crossed both D1R^Cre^ and Adora^Cre^ mice with a tdTomato reporter mouse (Ai9), and made coronal sections across the entire extent of the striatum and posteriorly including the amygdala complex (see new data in Figure 1D; Figure 1 —figure supplement 1 for D1R^Cre^, and Figure 3 —figure supplement 1 for Adora^Cre^). These images show, together with a re-analysis of our fiber positions, that we have targeted neurons in the broader area of the ventral tail striatum (vTS), but not in the more narrowly defined, and more ventrally located "AStria", as suspected by reviewer 1. Therefore, in the revised version, we do not refer any more to the "AStria", but rather, name the brain area studied here the more broadly defined "ventral tail striatum" (vTS).

Furthermore, we have re-written the end of the Introduction to introduce the posterior striatum more carefully (p.4, bottom, p. 5 top), and we now briefly discuss the targeting of the vTS versus the AStria (see Discussion, p. 19, top). Moreover, we now provide the fiber placement images of all mice in two new Supplementary Figures (see Figure 5 —figure supplement 1, and Figure 6 —figure supplement 1). Please also see our answers to the corresponding individual points of the reviewers.

3) Reanalyze the imaging data in accordance to the comments from Reviewers 2 and 3.

The imaging data has been completely re-analyzed, using "Caiman" software and the built-in deconvolution software in Caiman. Please see our responses to the specific points of the reviewers 2 and 3, below.

4) Address the reviewers' concerns related to the AP deconvolution method employed by the authors.

This concern was addressed by completely re-analyzing the Ca-imaging data (see point 3 above), and by switching from our custom-written deconvolution analysis to the built-in deconvolution analysis of Caiman; the latter is based on previously published papers (Pnevmatikakis et al. 2016 *Neuron*; Giovannucci et al. 2019 *eLife*). Please see our detailed response to the corresponding individual points of reviewers 2, and 3.

5) Include important missing controls as highlighted by the reviewers (e.g., positive control for optogenetic stimulation experiments, negative control for the rabies tracing data).

Additional experiments have been performed for both points.

First, for the optogenetic stimulation experiments, we have added additional data with longer blue light pulses for the Adora+ neurons. The new experiments further confirm the results in Figure 4, showing that vTS neurons do not directly modulate movement – or freezing of the mice (see new Figure 4 —figure supplement 1).

Second, we have performed additional control experiments for the rabies virus constructs (new Figure 7 —figure supplement 1).

6) Provide a rationale for why only male subjects were included in the study. Specifically, the authors should: a) state the sex in the abstract, b) explain why only one sex was used in the methods, and c) acknowledge and discuss the limitation of the exclusion of one sex in the discussion.

We have now stated the sex of the investigated animals in the abstract, and we have explained the choice of male mice in the Methods (new text in p. 28, top). Furthermore, we have discussed possible sex-specific differences regarding the role of the vTS in fear learning (p. 22 bottom, p. 23 top).

7) Demonstrate that the Cre lines used are valid as tools to achieve genetic access to the neuronal populations of the AStria.

The use of D1R^Cre^ and Adora^Cre^ mouse lines, to target direct- and indirect neurons/MSNs of the striatum, has been previously documented in many studies, albeit focusing on the dorsal striatum. For the vTS and including the AStria, a recent study has documented in detail the distribution of D1R+ and D2R+ neurons (Gangarossa et al., 2019; cited in our paper). We have added new anatomical data which shows the spatial distribution of Cre-expressing neurons in the D1R^Cre^ mice (Figure 1D; Figure 1 —figure supplement 1), and in the Adora^Cre^ mice (Figure 3 —figure supplement 1) in the vTS and AStria. This data is in line with the previous findings of Gangarossa et al. 2019 regarding the spatial distribution of D1R+ and D2R+ (in our case, Adora+) MSNs in the vTS and AStria. The data furthermore shows that there are very few neurons in adjacent cortical- and amygdala areas that would give rise to Cre-dependent expression in the Cre mouse lines. On the other hand, this data has revealed that Cre – expression in the D1R+ mice does not fully reach the ventrally located AStria (see Figure 1 —figure supplement 1C). This finding supports our decision to now name the targeted brain area ventral tail striatum (vTS) and no longer "AStria" (see above, point 2). Please also note that the new data in Figure 1 —figure supplement 1 and Figure 3 —figure supplement 1 show that D1R+ – and Adora+ neurons project to the entopeduncular nucleus / GPi and to the GPe, respectively, hallmarks of the connectivity of direct- and indirect pathway neurons in the Striatum. Taken together, with these additional experiments, we are confident that the Cre mouse lines used here allow selective access to D1R+ and Adora+ neurons of the direct and indirect pathway in the vTS.

8) Please ensure your manuscript complies with the eLife policies for statistical reporting: https://reviewer.elifesciences.org/author-guide/full "Report exact p-values wherever possible alongside the summary statistics and 95% confidence intervals. These should be reported for all key questions and not only when the p-value is less than 0.05." This should be reported in the main text.

In the revised version (as well as in the first version), we include detailed statistical reporting. We have double-checked with the reporting guidelines of *eLife* and, to our best knowledge, we comply with them.

9) Please include a key resource table in your methods and clarify whether animals were tested in the light or dark cycle.

A Key Resource Table has now been included. Also, we now state that mice were tested in the light phase (p. 29, top).

Reviewer #1 (Recommendations for the authors):Thalamic (PIN/PIL) and cortical regions (TeA/Auv) are missing or only weakly present when the connectivity of Astr is described. Romanski and Ledoux (1993) showed strong TeA-Astr connectivity, and Ledoux et al. (1990) as well as Barsy et al., (2020) highlighted a strong thalamo-Astr projection from the area of PIL and SG. Furthermore, the latter study also investigated the CS, US and CS/US responsiveness of Astr neurons which were found to be (at least, partially) mediated by the thalamic inputs.

The reviewer refers to previous studies which showed evidence largely from anterograde labeling studies, for auditory-related inputs to the AStria / vTS, e.g. from the temporal association areas (Romanski and LeDoux 1993) and from higher-order auditory thalamic nuclei (LeDoux et al. 1990 and Barsy et al. 2020). We, on the other hand, have investigated the presynaptic inputs to the vTS in a different approach, using rabies-mediated backlabelling techniques specific for D1R+ – and Adora+ neurons in the vTS. We find many brain areas presynaptic to either D1R+ or to Adora+ vTS neurons (Figure 7G and 7H, respectively). Of note, the temporal association area ("TeA"), and the auditory cortex ("AUD") were also revealed in our dataset (Figure 7G, H).

Furthermore, we have looked again at the data, and found that back-labelling was indeed also present both in the PiL and in the PoT, two thalamic structures located close to the auditory thalamus, and which have been shown to process auditory- and multimodal sensory information (see references in the Results). We have now removed this data from the previous "Others" categories, and have added them separately under "PoT" and "PiL" in Figure 7G, H (see also corresponding changes in the Results text; p. 15 bottom, p. 16 top).

The LA-Astr projection seems to be also underestimated in the view of the earlier publications (eg. Jolkkonen et al., 2001).

Please see our response above for the general reasoning of whether data from the rabies approach is directly comparable with previous studies largely based on anterograde labelling.

More specifically, neurons in the BLA are present in high density in our analysis (see Figure 7G, H; black bars), especially for Adora+ neurons.

One of the reasons causing these connectivity differences could arise from the targeting of Astr. Identification of Astr territory is not trivial, since no marker is available which could differentiate AStr from the rest of the caudal (tail) striatum. Still, as earlier studies showed that these two striatal regions may receive distinct inputs (eg. MGN to the tail, while PIN/PIL to the Astr; LA mostly targets Astr and not the other parts of the caudal striatum), it is important to investigate the same and consistent striatal region. Paxinos mouse brain atlas indicates AStr from bregma ~-0.9 (AP) between LA and LGP (and more caudal, the stria terminalis) and right above the CeA. The dorsal border is uncertain, but mostly indicated ventral to the top 'corner' of the LA (like in a recent review by Valjent and Gangarossa, 2020). However, many representative images and schematic drawing highlight more dorsal and anterior regions to be targeted which resemble the ventral part of the striatal tail, rather than Astr. While most of the foot shock and CS responsive cells were located in the posterior part of the examined regions (more posterior from bregma -1; Figure 1-3), some injection sites (AAV, RV) and fiber optic locations were more anterior. Indeed, the authors also use the tail of the striatum term. As no comprehensive functional study is present (to my best knowledge) which simultaneously investigate the tail of the striatum and Astr, it could cause many discrepancies if the two regions are mixed.Thus, some of the discrepancies could have arisen from targeting the tail of the striatum rather than Astr.

We agree with the reviewer, who points out the difficulty in demarcating the AStria / ventral tail striatum (vTS). Because of the additional concern whether the Cre mouse lines used here are well-suited to drive Cre-expression in the AStria / vTS ("Essential revision point #7, above, and reviewer 3, point 2), we performed further anatomical experiments. For this, we crossed both D1R^Cre^ and Adora^Cre^ mice with a tdTomato reporter mouse (Ai9), and made coronal sections across the entire extent of the striatum and posteriorly including the amygdala complex (see new data in Figure 1D; Figure 1 —figure supplement 1 for D1R^Cre^, and Figure 3 —figure supplement 1 for Adora^Cre^). These images show, together with a reanalysis of our fiber positions, that we have targeted neurons in the broader area of the ventral tail striatum (vTS), but not in the more narrowly defined, and more ventrally located "AStria", as suspected by the reviewer. Therefore, in the revised version, we do not refer any more to the "AStria", but rather, name the brain area studied here the more broadly defined "ventral tail striatum" (vTS).

We hope that the additional anatomical data, and the re-naming of the brain area targeted in our study (vTS), have solved this issue.

Several recent (Barsy et al., 2020, Gilad et al., 2020; Talyor et al., 2021) and earlier studies (Bordi and Ledoux et al., 1994a,b; Apergis-Schoute et al., 2005; Han et al., 2008; Weinberger, 2011, etc) suggested that CS-US association and thus, neuronal plasticity can occur prior to the amygdala, in non-sensory (higher-order) thalamic regions, like PIN/PIL. Therefore, the following statement is only partially correct. "In the lateral (LA)- and basal amygdala (LA and BA respectively), auditory (CS) and somatosensory (US) information are integrated.."

We agree with the reviewer. This part of the Introduction has been re-written, to make the statements about the LA and BA more general (see p. 3, bottom).

Was there any level of dorsoventral topography found in the location of the Cs and/or US responsive cells (striatal tail vs. Astr)?

In response to this question, we now show the reconstruction of imaged neurons also in coronal planes (see Figure 2 —figure supplement 1 for *in-vivo* ca^2+^ imaging of D1R+ neurons, and Figure 3 —figure supplement 3, bottom, for Adora+). For the US- and CS-responses in D1R+ neurons (but not for the movement-ON responses), it appears that neurons located more ventrally have smaller responses. Nevertheless, such a topography would have to be investigated in more detail, and with a larger sample size. Please note that the neurons which are localized more ventrally in Figure 2 —figure supplement 1, are actually *also* located more anteriorly (see the "horizontal" view in Figure 2A), thus precluding a simply analysis merely in the dorso-ventral plane. Furthermore, please note that the apparent localization of these cells in the "BLA" as suggested in the coronal view (Figure 2 —figure supplement 1), is actually an artefact of the projection (compare with the "horizontal" view in Figure 2A).

Is it possible that the examined and responded D1 and Adora+ cells are actually located in two distinct striatal regions? It is discussed that these are rather topographically non-overlapping populations.

Indeed, in the previous version we had reported (p. 13, l. 292 – 293 of the original ms), that Adora+ neurons with movement-ON responses might be localized more anteriorly, than D1R+ neurons with movement-ON responses. However, because it seems difficult to back-up this observation with a statistical analysis, we have now removed the statement.

Is it possible to follow the Ca-activity of individual cells throughout the entire learning paradigm? If yes, plasticity of a single cell could be analysed (like in the thalamus, Taylor et al., 2021).

In response to the criticism by reviewer 2 and 3, we have re-analyzed the Ca-imaging data. The analysis is now based on a CNMF-E based approach to detect ROIs (in "Caiman"; see Methods and response to reviewers 2 and 3), and with this approach it should in principle be possible to register cells over various days. Unfortunately, the number of cells that can be registered successfully across days is quite low. We have therefore refrained from further analyzing the imaged neurons in a day-by-day fashion.

As Figure 7 showed some level of topography in the InsCx-Astr connectivity, were different sectors of InsCx targeted in the D1 and Adore-Cre animals used for patch clamp recordings in Figure 8?

As the reviewer remarks, there is an apparent difference in the density along the a-p axis of cortex of presynaptic neurons proving input to D1R+ versus Adora+ neuron (Figure 7I, J).

For the patch-clamp recordings, we did not target different areas but instead, we always targeted the posterior insular cortex (pInsCx) for the injections of AAV8 driving the expression of Chronos, (see Methods, p. 25, bottom).

We also clarified in the Results and Discussion that for this experiment, we targeted the pInsCx and not the S2, but that some spill – over of Chronos-expressing virus into the S2 cannot be excluded (see p. 16, middle, and p. 21 middle).

Pg 17. Between the AAV and RV, three weeks is indicated in the text, but 4 weeks in the figure.

Thank you for pointing out this inconsistency, which has been corrected (see p. 15, top)

Figure 8, Figure suppl. 4 B and C. It seems that distinct parts of striatum are targeted.

It is true that in the example images, the injection into the Adora^Cre^ mouse was targeted somewhat more dorsally than in the D1R^Cre^ mouse.

These images were taken from the control groups for D1R^Cre^ / Adora^Cre^ mice expressing eGFP, from the datsets in Figure 5 / 6. Because the behavioral experiments were performed with bi-lateral fiber placements (to ensure sufficiently high effect sizes) and bi-lateral expression of Arch (for the 'effect' group) and eGFP (control group), we think a certain amount of variability in the targeting is un-avoidable. This, however, should not have affected the validity of these mice as control mice.

In response to the criticism, we have now removed this figure supplement, since the anatomical analysis had remained somewhat preliminary. Furthermore, for the discussion, this data is not needed, and we feel that the targeting of the vTS in the silencing experiments of Figure 5 and 6 is now amply documented in the new figure supplements to Figure 5 and Figure 6.

As there is very little data about Astr, it would be interesting to examine the basic anatomical organization of Astr. For example, the proportion of D1+ and Adora+ cells, the presence of other types of neurons (cholinergic, PV cells, etc). But, these may be far beyond the scope of the present study.

As the reviewer noted, we think these additional ideas for studying anatomical properties and interneuron types of the vTS goes beyond the scope of the current paper.

Reviewer #2 (Recommendations for the authors):It is disappointing that the presentation and some of the analyses do not do justice to the work. Please see my points below. In particular, the heavy handed analyses of the fluorescence transients is too much and fraught with potential for error. More importantly, it obscures what appear to be large differences that could be extracted more easily (and convincingly). Please see the "approach section below". Lastly, at times the authors interpret their data and results too optimistically.

Please see our responses to the specific points below.

Specifically, the authors should:1. Reanalyze the data using only z-score F(t).

In response, and also to satisfy the criticism of reviewer 3 (point 4 and 5), we have newly analyzed the Ca-imaging data, from scratch. First, for the detection of ROIs (see reviewer 3, point 3), we have used the analysis routines provided by Caiman. See new display of Caimaging data in Figure 1H – R, and in Figure 3E – O, which are now based on Z-score instead of on deconvolved event frequencies.

Regarding the analysis of Ca event activity during the four different conditions (combinations of movement or freezing with presence of absence of CS; "conditional event frequencies"), we continue to think that a read-out of discrete events is advantageous over simple z-scored Ca-traces. Nevertheless, we now use the previously described Ca-deconvolution approach available in the Caiman software, based on the previous papers by Pnevmatikakis et al. 2016 Neuron, and Giovannucci et al. 2019, *eLife* (both papers are cited in the manuscript). The complete re-analysis of our ca^2+^ imaging data should address possible caveats that might have been present in our previous custom-written deconvolution analysis.

2. Show more primary data – images of cells, etc…

We now extensively show color-coded z-scored fluorescence traces aligned to different stimuli (tones, footshocks) / behavioral variables (movement – ON)(Figure 1H-R; Figure 3 EO).

3. Refrain from reaching conclusions too early in the results especially that it is the contribution of plasticity of the neurons that is being studied in the silencing experiments (as opposed to the need for activity).

We have revised the text on all occasions in which we implicated that silencing during the footshock should affect a "plasticity". Rather, we now use the term "footshock – driven activity" or similar in these instances (see Results on several occasions). Nevertheless, in the discussion we express our interpretation that footshocks likely drive plasticity in these neurons (p. 19, middle – bottom).

4. Improve the presentation of the methods

The Methods for the analysis of Ca-imaging data has been re-written, according to the new analysis we have performed (see Methods, pages 31 – 33).

Please read the detailed comments below.Abstract:1. Imprecise wording. The notion of "balancing" is unclear in "defensive behaviors need to be finely balance" which makes it hard to understand exactly what is being studied.

We added "in the presence or absence of a threat-predicting cue" (p. 2, l. 17), to make it clear that we mean the balancing of learned freezing behavior across the CS- and no-CS periods.

2. "In-vivo optogenetic silencing during the training day showed that plasticity in D1R+ AStria neurons contributes to auditory-cued fear memories"Please see my comments on the interpretation of the silencing experiments.

This sentence was modified as pointed out above (major point 3), using the term "footshock-driven activity" instead of "plasticity" (p. 11, l. 254 – 255).

Results:1. Show some images of neurons with corresponding fluorescence transients to understand the quality of the data. I can't find them anywhere in the figures.

We show images of neurons (Figure 1C), and raw traces of ca^2+^ (Figure 1 E-G).

2. Line 111 – "Seemed to increase". What does this mean?

This statement has now been deleted, when re-writing the Results part to accommodate the newly analyzed Ca imaging data.

3. Line 114 – Claim to have derived APs by deconvolution.

This sentence has now been removed, because – as suggested – we simply analyzed most of the Ca imaging data in the form of z-scored Ca traces.

4. Line 127 – "Similarly suggested an increased".

We feel this statement made sense in the framework of the previous analysis. An increase in the tone response throughout the population of neurons is also present in the z-score traces in the new analysis (see Figure 1 I, J, K). However, since this sentence related to the previous analysis based on Ca-event frequencies, it was now removed.

5. See comments on methods – by the time we get to summary data as I Figures IJ-L it is very hard to understand what these mean. They are in au units x 10e-4. What does this mean? If they are au then they can be remapped.

This data is now shown in the form of aligned z-scored Ca traces (Figure 1 H-J), and thus use the unit of z-score.

6. Line 154 "These experiments thus show that D1R+ AStria neurons increase their AP firing activity during the onset of movements after a period of immobility. "The analysis in this section is potentially problematic as tones are being played and the animal is starting and stopping movement. It is not clear that the response is to the movement change as opposed to the tone. Please show the histogram of tone start and stop times relative to the movement onsets.A better analysis is one that models the contributions tones, movement starts, stops, and US. This can be done with a GLM and will alone one to disambiguate these confounds.I am especially worried because of the follow-up sentence "Line 155 were more pronounced during the CS than in the absence of a tone" which suggests that these are not movement cells.

In the course of the re-analysis of the data (Ca Z-scores instead of deconvolved Caevents), we have now split the analysis of movement – ON responses in two blocks. First, we analyze the movement – ON responses for the "no – CS" periods, when no tones were given (Figure 1 M-O for D1R+ neurons, and Figure 3J-L for Adora+). This analysis unambiguously shows the presence of movement – ON responses. Second, we have then aligned the z-scored Ca traces to the onset of movements *during the CS*. As requested by the reviewer, we have now computed the histograms of the times of CS occurrence relative to the movement – ON transition (Figure 1 —figure supplement 3 for D1R+). This indeed shows, as the reviewer suspected, that part of the movement – ON responses were preceded by tones.

In order to correct for these responses, which might have been caused by tones rather than by movement – ON transitions, we removed all Ca – traces that were preceded by 0 – 400 ms by a tone, leading to the removal of about half of all traces (see Figure 1 —figure supplement 3, numbers given at the bottom). The corrected traces are shown in red in Figure 1 —figure supplement 3. The corrected traces still show a substantial increase of Ca in D1R+ neurons during movement – ON transitions *during the CS* (Figure 1—figure supplement 3, red traces). For the analysis of the number of neurons responding to movement – ON transitions during the CS (Figure 1S, filled data points), we used the corresponding tone-event-corrected traces.

In summary, responses to movement – ON transitions are clearly present in D1R+ neurons, and in Adora+ neurons (see the analogous analysis for Adora+ neurons in Figure 3M-O, and Figure 3P). Our new analysis furthermore shows that the number of neurons that respond to movement – ON transitions is larger in the presence of a CS than in its absence (Figure 1S, and Figure 3P). However, because of the caveat that part of the movement – ON responses are possibly caused by preceding tones, we have removed the statements which described a positive interaction between "CS" and movement – ON responses (see Results).

7. Line 159 – The conclusion in this sentence cannot be justified based on the data shown so far.

This sentence about a non-linear interaction of a representation between tones and movement – ON responses was removed. See also our response to point *[8]*.

8. Line 168 – One cannot make these conclusions by comparison of event-triggered averaging.

This sentence was also removed; in relation to point *[8].*

9. Overall Figure 1 is very hard to follow. The number of panels should be reduced, labels should be placed to show which analyses are relative to movement, CS, etc…

We have strived to make Figure 1 easier to follow by the various changes that went along with the re-analysis of the data (Z-scores). As suggested, we have also introduced labels to indicate the principal response types (US, Tone, movement); similar changes were done in Figure 3.

10. Figure 2 is very nice.

Thank you; this Figure is now based on newly analyzed data, since both the coordinates of cell maps, as well as the response strength of the cells was changed slightly after the reanalysis of the Ca imaging data.

11. The paragraph starting on line 252 is confusing. Are the authors saying that the # of cells responding goes up (i.e. response fidelity) but that the response per active cell (estimated AP content) does not? Both the positive and negative conclusion refer to Figure 3R. I can't find a definition of response fidelity, which is used several times.

Sorry for the confusion. In the revision, we now removed the analysis of "response fidelity", so this should no longer be a concern.

12. The analysis of Figure 3S says that significance was judged by a KS-test, which is usually used for comparisons of unbinned cdfs. The graph is an average +/- error bar. How was the KS test used?

This is likely a mis-understanding; we used a Kruskal-Wallis test (abbreviated as "KW"). As detailed in Materials and methods, the Kruskal-Wallis test is a non-parametric version of a one-way ANOVA. This test was used to test the "general" significance of changes according to time (three days) and class of events (four combinations of movement / freezing and CS / no-CS), before using post-hoc comparison (Dunn's test). In the revised version, we always write out the test name for "Kruskal-Wallis" to avoid a possible confusing with "KS".

13. If the freezing habituation data in 1S and 3S cannot be analyzed statistically, it should be removed from the panels as one cannot help but interpret the data shown quantitatively.

After the complete re-analysis of the ca^2+^ imaging data, we chose to keep the data (see Figure 1 U, 3R).

14. The Venn diagrams in 3V and 2D would benefit by labeling the numbers in the main intersection areas.

This was done accordingly, albeit only for the "simple" overlaps between tone- and movement – ON responses to keep the graph light.

15. In figure 4, the baseline movement of the Adora2a and D1R groups is very different. Are the two genotypes equivalent? These are BACs so it would not be too surprising.

We think the absolute movement indices, which are analyzed by the "ezTrack" software from the videos of the mice (see Methods, p. 31, middle), should not be compared directly. For example, the movement of the Adora^Cre^ mice, in the newly added data (Figure 4 —figure supplement 1), shows much higher apparent levels of movement index. Indeed, the exact values depend on the calibration of the system, the camera used, and possibly on the size of mice. Thus, we refrain from deriving conclusions regarding possible differences between absolute movement indices.

16. For Figure 4, what in vivo proof is there that the cells are being activated? Given the negative conclusion, such a control is important.

We attempted optrode recordings to show that optogenetic stimulation of D1R+ neurons by blue light pulses *in-vivo* can drive AP-firing. Unfortunately, we were unable to record light-evoked activity with sufficiently high throughput, due to direct light -evoked artefacts in the recordings, and/or misplacements of the optrode (N = 4 mice, data not shown).

We therefore turned to a simpler approach, at least for Adora+ mice. In fact, in the original data set, we had used very brief light pulses (1 ms; 10 mW at the fiber tip), motivated by the fact that the kinetics of Chronos is fast (Klapoetke et al., 2014). We therefore validated in additional experiments, with N = 5 available Adora+ mice, whether optogenetic stimulation of Adora+ neurons in the vTS with longer light pulses (2 ms, and 5 ms; 10 mW), would produce comparable results to the ones in Figure 4. Indeed, these new experiments similarly did not show an effect on the movement of the mice (see new Figure 4 —figure supplement 1). The new experiments further corroborate our findings that optogenetic stimulation of Adora+ and D1R+ neurons in the vTS do not have a direct effect on the movement of mice.

17. The experiments motivated by the paragraph starting at Line 325 are very nice. However, the paragraph is troubling. The manipulation is to reduce activity and it should be phrased in that way. Instead, a hypothesis based on plasticity is presented, from which the experimental manipulation of activity is indirectly motivated.

The paragraph has been re-written, using the notion of "footshock-driven activity" instead of "aversively – motivated plasticity" of D1R+ neurons.

18. There are differences in baseline freezing (5F) before the light is turned on that are of similar magnitude (relative to the low level of freezing) as seen in 5H. Similarly, the comparison between no effect in 5G right to less effect in 5I left may not be fair given that the latter is in "steady state" and the former not. Is an RM-ANOVA the right way to go?

Thank you for your detailed observation of the data in Figure 5F. In Figure 5F (now Figure 5D in the revised version), and Figure 5H (5F in the revised version – note that below we only used the "new" Figure numbers), we show averaged freezing percentages across all mice in each group, at a time resolution of 10 s. The reviewer refers to "*differences in baseline freezing (5F) before the light is turned on*". In these experiments, 30s trains of tones (each 0.1 s, given at 1 Hz) are given (blue – shaded area), and then a final footshock is given immediately *after* each CS (Figure 5F, yellow vertical lines). The yellow light, to activate Arch, is given 1s before each footshock, for a duration of 3s. Thus, one would have to look at the last 1-2 freezing values before each yellow vertical line, to check for "differences in baselines freezing before the light is turned on". This would be at times when the CS is on (blue – shaded areas). Indeed, in the time-binned analysis, the freezing levels differ between the two groups especially during CS 4 and CS 5, but not in a consistent manner – once the eGFP (control) mice show higher freezing (CS5), and once the Arch mice show higher freezing (CS 4)(Figure 5E, left).

It should be noted that currently, there seems to be no consensus in the fear learning field whether the CS applied on the training day (during pairing with the US), already has acquired the properties of a CS for the animals. Thus, it is possible that freezing during the CS, on the training day, can be seen as a "prolongation" of an ongoing contextual freezing, that buildups over longer times during the training session (Figure 5D, green and black traces).

In the previous, and revised version we have analyzed the freezing in 30s time bins *preceding* each CS (lower grey bars in Figure 5E; 5F, *right*), and the freezing during each CS (Figure 5E, *left*). This data shows that both within the control- and the Arch mice, freezing is similar during the CS, and during the no-CS times (Figure 5F, compare data between *left* and *right* panel), which supports the view that during the training day, mice freeze in an increasing manner in response to the general context, but not specifically in response to the tones.

In summary, we think small fluctuations in freezing levels between the groups on the training day, and within CS4 and CS5 in this case, are not of biological relevance.

Please also note that, because we re-included N = 2 eGFP mice, and N = 3 Arch mice that had been previously excluded because they froze less than 20% at the end of the training day (see reviewer 2, point 26), the exact form of the traces and positions of datapoints in this Figure has slightly changed.

19. Similar concerns exist for the data in Figure 6.Can the differences in Figure 5 or 6 be used to identify individual mice as ones that received silencing and ones that did not? It seems unlikely, especially for Figure H. It might have been better to run cohorts of mice in which the silencing was only applied in the retrieval portion. Otherwise, it is unclear if the relatively modest effects are due to a difference during training that is not reflected in freezing rates.

For a general answer regarding to the "similar concerns" in Figure 6, please see our answer above for Figure 5.

The experimental approach here was to suppress footshock-driven activity in either D1R+ or Adora+ neurons of the vTS, and then observe whether this manipulation would impair fear learning. This experiment has revealed opposite effects when applied to neurons in the direct

(D1R+) and indirect pathways (Adora+)(a contribution to auditory-cued fear memory for D1R+ neurons), as opposed to an increased contextual fear memory component for the Adora+ mice (but see also comment below on the changed statistical significance in the Adora+ dataset). The reviewer proposes that "it might have been better" to silence the activity of the neurons during the recall day. We feel, however, that the experiment as run by us makes sense, and has produced interesting results, whereas silencing during the recall day will require further considerations as to when best to silence (since the effects of D1R+ and Adora+ neurons are produced at different times: CS versus no-CS times). Thus, we think that silencing the activity of D1R+ and Adora+ neurons during the tones / context phases of the memory recall day is beyond the scope of the present paper.

Regarding the point that it is "unclear if the relatively modest effects are due to a difference during training that is not reflected in freezing rates": The rationale of these experiments was to suppress footshock – driven activity in a specific neuronal subsystem, and by this to most likely reduce plasticity in this sub-system (we express this view in the motivation paragraph to the silencing experiments; see p. 11, middle – bottom). We find it encouraging that this manipulation, which presumably impaired plasticity in the vTS, leads to selective changes in memory recall one day later, which indicates that the vTS contributes to the formation of an auditory-cued fear memory. We don't think it is problematic that the manipulations on the training day did *not* impair the increasing wave of "contextual" or "anticipatory" freezing observed on that same day, because other brain areas might drive the contextual, or anticipatory freezing observed on the training day.

Changed statistical significance in the Adora+ data set:

As requested by this reviewer below (point *[26]*), we have now re-introduced the previously excluded mice (which were excluded based on the criterium that they showed less than 20% freezing at the end of the training session, spanning the times of the fifth and sixth CS-US pairing). While the statistical significance in the D1R+ dataset was maintained (see Results), the statistical significance in the Adora+ dataset was lost, while there was still a trend in the data showing an increase in the freezing at no – CS times on the fear memory recall day (p = 0.0512; two-way repeated measures ANOVA; previously, p = 0.041). Therefore, we have now toned down our conclusions regarding the Adora+ mouse, and merely speak about a "trend" in the data towards an increased freezing during no – CS times after suppressing footshock-evoked activity in Adora+ vTS neurons (see e.g. changed text in p. 14, l. 327).

20. The main text does not reference controls for the rabies experiments which generally should include (1) TVA dependence of rabies infection, (2) G dependence of spread, (3) Cre-dependence of the DIO/floxed constructs. It is important to do these with the same batch of viruses used for the experiments (ideally in parallel!).

We have now performed additional control experiments as requested by the reviewer (see new Figure 7 —figure supplement 1, and changed Results text; p. 15 top).

21. I hesitate to make too much of the differences shown in 7I/J given N=2, especially without a careful analysis of all the starter cells to show that the number and distribution of these were the same across genotypes.

In response, we have toned-down our conclusions about this experiment, and now merely report the observed distribution (see p. 16, top).

22. The currents in Figures 8G-H are huge and loss of voltage clamp must have occurred, especially when measuring NMDA receptor currents.

Yes, the optogenetically evoked EPSCs are larger than most reports in the literature for long-range excitatory connections in the forebrain.

We have extensive experience with imposing voltage-clamp also on large and fast EPSCs. This experience dates back from our work at the calyx of Held synapses (where EPSCs can be up to 20 nA; see e.g. Meyer et al. 2001 J. Neuroscience; Kochubey et al. 2009 J. Physiology). Also, at the MNTB – LSO inhibitory connection, we more recently reported optogenetically-evoked IPSCs of up to 20 nA (Gjoni et al., 2018 J. Physiology). This requires to minimize the series resistance (Rs). Also, the finding that the optogenetically-evoked EPSC at the pInsCx – vTS connection are surprisingly large, is a sign that our voltage-clamp conditions might be appropriate, because imperfect voltage-clamp would lead to an *underestimation* of the true synaptic conductance (see discussion in Gjoni et al. 2018 – in this paper we reported significantly larger conductance values for IPSCs than those reported before at the same connection). For these reasons, we think we are well-positioned to report the occurrence of unusually large EPSCs.

We now discuss that the cortical inputs to the vTS are of surprisingly large amplitude (p. 21, middle).

Approach:1. Large lesions even for optogenetics because of use of 1.25 mm cannula.

This seems to be a misunderstanding. The 1.25 mm outer diameter ceramic ferrules remain outside of the skull. The optic fibers are inserted into the brain, and have an outer diameter of 230 µm (see also improved methods description, p. 30, bottom).

2. Line 763 -- "If mice from any experimental group showed low freezing, they were excluded from the analysis; we used a threshold of 20% time spent freezingduring the fifth and sixth CS-blocks on day 2 (7 out of 59 mice for the behavioral experiments were excluded)."It is good of the authors to give the exclusion criteria. However, this is worrisome. Some of the loss of function experiments are based on the hypothesis that activity is necessary for learning. Therefore, this discards the poor learners, right?What is the breakdown of exclusion across the different experimental and genetic conditions? Why not simply include all animals to capture the natural variability?

We have re-added the previously excluded mice to the datasets in Figure 5 and Figure 6. Corresponding changes have been made in the Figures, and in the Results text. Please also see our answer to your points [20], [21] above.

3. The deconvolution approach is worrisome for a few reasons.a. Ca entry into these neurons is highly non-linear with single AP, burst AP, up-state APs, and GPRC-mediated influx. It is unclear that one can use a linear model of # of APs to [Ca] (to F(Gcamp6m)). It does not seem that this step is necessary (especially given the heavy filtering).b. Data is acquired at 10Hz but then a band-pass filter at 2-4 Hz (If I read it correctly although the methods are not very clear). This is much less than the Nyquist limit. Why? Also why place a low-frequency limit (i.e. why use a band pass as opposed to a low-pass to simply get rid of shot and electronic noise?).c. The deconvolution kernel is arbitrary and uses a rise time that cannot be captured with the 2-4 Hz band pass. The authors need to show that 0.5 s tau(decay) is justified. There is no justification for choosing an amplitude of 2.d. Line 793 – The phrase "deconvolution yielded a… proportional to AP firing rate" must be removed as it cannot be justified.e. The deconvolved signal (R) is further filtered with a box car – Why? The filtering is done upstream on the fluorescence. It should not be done again.f. "local peaks" exceeding a threshold (arbitrarily set to 2) were detected using a "first-derivative" method. Why add a threshold here? It seems unjustified. Second, how does one detect a peak with a first-derivative? Shouldn't it be a second derivative? Or are they detecting fast-rate of rise, in which case why set a peak amplitude threshold?4. All of these concerns are acknowledged later (line 801) in "we did not aim to infer exact AP spiking rates of cells". Then why do this and present it in this way?5. They should simply get rid of all these steps and analyze the z-score raw F (calculating a DF/F and then z-scoring is no different than just z-scored the Z).6. y comparing the7. The section starting on line 809 indicates that different baselines were chosen for comparison for different types of cells. This introduces a circular bias. One needs to be able to statistically show that a cell is a "move-on" or CS responder without changing a baseline. Why not use the Z-score values in the time-bin without picking a base-line – i.e. compare activity in that window to all activity?8. Line 798 – The amplitudes of the events from deconvolution (referred to as estimated AP content, EAC [a.u./s]) were proportional to the amplitude of the ca^2+^-transients (e.g. see Figure 1 —figure supplement 1E).a. I can't find the referenced data in a figure. The sentence suggests a scatter plot with some kind of regression analysis?b. If this is true, why do it all? Just use the F as suggested above and save all these potentially problematic analysis steps.

Thank you for your detailed comments on our previous deconvolution – based analysis of Ca transients and underlying neuronal activity. As suggested by this reviewer and by reviewer 3, we have now completely re-done the analysis of the Ca imaging data. As requested, we have performed most analysis steps on the Z-scored fluorescence (Ca) traces (see new panels Figure 1H-R and Figure 3E-O for D1R+ and Adora+ neurons, respectively) (please note that the Caiman software does not allow access to deltaF/F traces so we used Zscored traces). The fluorescence data were newly extracted using an CNMF-E based approach.

We only maintained a deconvolution – based approach for the analysis of "conditional frequencies" of Ca events in panels Figure 1U, and Figure 3R. For this, we used the built-in deconvolution approach of Caiman, instead of the previous custom-written analysis. Thus, we feel that all issues raised by the reviewer in these above comments, relating to filtering and other technical issues, should have been resolved by the new analysis of the data.

Please see the new description in Material and methods, which succinctly describes how we analyzed the Ca data (p. 32 – 34). Please also see our response to the related points 3 and 4 of reviewer 3.

Reviewer #3 (Recommendations for the authors):1. The study used only male mice. There is no basis for excluding females.

We have now stated the sex of the investigated animals in the abstract, and we have explained the choice of male mice as a model in the Methods (new text in p. 24, top). Furthermore, we have discussed possible sex-specific differences regarding the role of the vTS in fear learning (p. 22 bottom, p. 23 top).

2. The authors did not verify specificity of D1R and Adora lines (tdTomato + RNAscope for D1R and Adora). Without this verification, it is unclear if the fluorescent patterns reported actually reflect patterns of D1R and Adora neurons.

We have used a D1R^Cre^ mouse line (EY217 line from the GenSAT initiative), and an Adora^Cre^ mouse line (KG139 line from the GenSAT initiative), which both have been used widely by previous studies in the striatum. These mouse lines have been well characterized in the GENSAT project (see expression profiles of these two lines on the GENSAT website; http://www.gensat.org/cre.jsp). We see no obvious reason that any of the two lines should express outside of its previously documented expression domain.

In response to this question, and to Reviewer 1 point *[3]*, we have performed additional anatomical control experiments. We have crossed the D1R^Cre^ and the Adora^Cre^ mouse line with a Cre-dependent tdTomato reporter line (Ai14), and we have then analyzed the location of tdTomato-positive cells throughout the striatum and the amygdalar complex (N = 3 mice each). Representative images on three different a-p levels, after aligning with the Paxinos mouse brain atlas, are shown in Figure 1 —figure supplement 1 and Figure 3 —figure supplement 1.

For the D1R^Cre^ x tdT mice, it can be seen that tdTomato-positive cells are present widely within the striatum ("CPu / CP in Paxinos), and that tdTomato-positive axons project to the entopeduncular nucleus / GPi (Figure 1 —figure supplement 1B, C), a hallmark for neurons of the direct pathway. Interestingly, as was also noted in the original paper describing these lines (Gerfen et al., 2013), tdTomato expression was sparse in the cortex and in the claustrum; moreover, we did *not* observe tdTomato expression in the intercalated cell masses that surround the BLA. This feature of the D1R^Cre^ (EY217) line is advantageous for our purpose, because it ensures that we did not inadvertently image, or silence D1R-expressing neurons in adjacent structures to the vTS (like deep layers of cortex, claustrum, intercalated cell masses surrounding the BLA). Indeed, in another D1R^Cre^ line from GENSAT, expression that matches more accurately "the full pattern of endogenous expression of the Drd1a gene" was observed (line FK150; citation from Gerfen et al. 2013). In that line, expression is also seen in cortex, claustrum, and the intercalated cell masses (see images on the GENSAT website; http://www.gensat.org/cre.jsp). One possible dis-advantage resulting from the choice of the D1R^Cre^ (EY217) line is that expression more posteriorly and ventrally becomes weak in this mouse (see Figure 1 —figure supplement 1C). In this respect, it also seems more cautious that we now refer to the neurons targeted in our study as ventral tail striatum (vTS), but no longer as "AStria" (see also Reviewer 1, point *[3]*).

For the Adora^Cre^ x tdT mice, it is again seen that tdTomato-positive neurons are contained throughout the striatum as expected, and axonal projections target the GPe (Figure 3 —figure supplement 1A, B, C), a hallmark of the indirect pathway. TdTomato-positive neurons are found in the CPu/CP extending partially into the AStria. Only few Cre- positive cells are found in structures adjacent to the CP/CPu and AStria. These data show that this mouse is well-suited to target neurons of the indirect pathway (D2R+ or as here, Adora+) in the vTS.

Taken together, the present and previous characterization of the two Cre mouse lines used here is quite extensive, and we do not expect that the proposed control experiment (RNAscope for D1R, Adora, and tdTomato in each mouse line) would reveal significant weaknesses of the used lines. Moreover, our slice electrophysiology data shows that input EPSCs from the pInsCx/S2 received by D1R+ and Adora+ neurons are of different amplitudes, different E/I ratios, and show different AMPA/NMDA ratios after fear learning (Figures 8, 9). This again confirms that we are dealing with two separate neuronal populations. All this evidence suggests that the DR1^Cre^ and Adora^Cre^ mouse lines used here are valid tools to target neurons of the striatal direct- and indirect pathways.

In response to this criticism, we now show images of the D1R^Cre^ x tdT and Adora^Cre^ x tdT mice in Figure 1 —figure supplement 1 and Figure 3 —figure supplement 1, to introduce the expression pattern of each Cre mouse line in the vTS.

3. Inspection of the heat plots shown in figures 1 and 3 indicate the fluorescent data contains repeating 'neurons'. For example in Figure 3, the two signals in rows 10 and 11 are identical. This is particularly notable because the repeating 'neuron' shows the largest change in fluorescence. In addition, rows ~17 and ~34 appear virtually identical as well. This is a major problem and indicates the authors have an issue in their analysis pipeline. Greater care needs to be taken to ensure repeating neuron's are not reported and analyzed.Same issue occurs in Figure 1. Rows 3 and 4 are identical signals. Repeating 'neurons' appears to be a problem throughout.

Thank you for this detailed observation of our data, this has previously escaped our attention. With the re-analysis of all imaging data in Figure 1-4, this problem should be solved. For the previous analysis, we had hand-drawn ROIs around active cells. Since we imaged over three focal planes, apparently some cells had been included twice in the analysis. For the new analysis, we used a CNMF – E based approach in the "Caiman" software to identify active cells, and this software makes a co-variance test between cells and then excludes "repeating neurons", so the problem should not persist in the new data set.

4. GCaMP6m is a fluorescent calcium indicator. While calcium entry into neurons is required for AP-driven vesicular release, calcium entry into a neuron can reflect many other processes. In this manuscript, the authors wish to examine single unit activity by acquiring and deconvolving fluorescent signals. In fact, there are known issues relating fluorescent signals to action potentials in the striatum:https://www.biorxiv.org/content/10.1101/2021.01.20.427525v2.fullIf the authors wish to analyze spiking activity of the striatal neurons, they need to use techniques that directly record action potentials. These techniques are readily available. If the authors wish to record fluorescent changes resulting from calcium influx, they need to embrace this decision and only analyze changes in fluorescence data.

Thank you for pointing out the paper Legaria et al. 2022 (Nat. Neuroscience). In it, the authors show that measuring global Ca averaged over potentially many 100s of neurons in fiber photometry, correlates only marginally with the AP-firing activity of striatal neurons. The authors then additionally use miniature microscope Ca imaging with similar methods as used here (their Figure 3), to differentiate between the soma- and non-somatic Ca signals. Thus, it is clear that miniature microscopy has a big advantage (=cellular resolution) over fiber photometry.

This said, we have followed the advice of this reviewer and reviewer 2 and have completely re-analyzed the Ca – imaging data. We now have used CNMFE-based methods for ROI (cell body) detection as implemented in Caiman, and we have expressed most results in terms of zscored Ca traces (see new panels of Figure 1G-Q, and Figure 3E-O). We still use deconvolution of Ca-traces for the final analysis of "conditional Ca-event frequency" (see Figures 1T, 3R), because for this analysis it was necessary to count events. We now use a deconvolution analysis in the Caiman package, based on previously described methods and shared analysis programs (Pnevmatikakis et al. 2016 *Neuron*; Giovannucci et al. 2019 *eLife*).

Taken together, we hope that the new analysis of fluorescence (Ca) data in Figures 1 – 3 has improved the logical flow of the presentation of the Results. We have also dropped all statements in the text in which we had suggested that Ca-deconvolution can derive "APfiring".

Please also see our answer to "Essential revision points 3, and 4".

5. I found it very difficult to track the logical flow of the analyses. The authors start off by showing 'neurons' are responsive to cues. But then almost immediately pivot to showing they do not in fact respond to cues, but movement during cues. A lot of figure space and analysis is devoted to this, such that the big picture is lost. I would find it more convincing to develop analyses that directly compare cue vs. movement vs. cue x movement responding from the outset. A regression approach may be useful. Take the 30 s prior to cue and 30 s cue. Have one regressor be cue on vs cue off, another be movement on vs movement off and the last be the intersection of cue on/off and movement on/off. This would simultaneously compare each regressor to each 'neurons' activity pattern, determining which best captures change in fluorescence.

Thank you for this feedback. We admit that the previous version of Figure 1 was dense, and the data as presented following the deconvolution analysis was difficult to grasp. We have attempted to improve these issues of data presentation by the new analysis of the data (largely based on Z-scored fluorescence traces, see above), and by using improved organization and labels in Figures 1, and 3.

Nevertheless, the fundamental properties of the data remain the same as before; i.e. there are both responses to sensory cues (US, and CS – driven responses), as well as responses to behavioral state variables, like movement – ON transitions of the animals. We have tried to improve the logical flow, by now clearly showing movement – ON responses in the absence of tone (CS) stimulation *first* (Figure 1M-O; Figure 3J-L). Nevertheless, within the CS blocks (tone stimuli every s), the issue persists that movement – ON transitions can occur superimposed with tones. Indeed, a re-analysis of the occurrence of the CS (tone) events relative to movement – ON transitions shows that in a significant number of movement – ON transitions during the CS (~ 50% for D1R^Cre^ mice; see Figure 1 —figure supplement 2), the movement – ON transitions were, in fact, preceded by 0 – 400 ms by a tone (see also our answer to reviewer 2, point [8]). This phenomenon was more pronounced in the D1R^Cre^ mice, and likely indicates that tones can sometimes induce the onset of movements, with a delay of ~ 200 – 400 ms. We feel that in this situation, a generalized linear model (GLM) would be of limited usefulness, because tone- and movement – driven events can sometimes occur with essentially similar delays, and in a similar periodicity (the periodicity is imposed by the periodic tones during the CS block). Thus, we applied a timing – based analysis, in which we removed the Ca responses for those events in which the movement – ON transition happened to be preceded, by 0 – 400 ms, by a tone. The resulting "corrected" average Ca transients showed a similar overall trend as the non-corrected traces (Figure 1O-Q, compare red and black traces; similar for Figure 3 M-O for Adora^Cre^ mice).

Therefore, we feel that have fully addressed the possible concern of the overlay of movement – ON responses, and tone responses. Because of the possible triggering of a part of the movement – ON responses by tones (see above), we now removed all statements which had claimed a non-linear interaction between movement – ON and tone responses (see Results, and Discussion).

6. Do not show hypothetical behavior data (as in Figure 1A). Only show real behavior data.

We have now changed the scheme of Figure 1A (and of Figure 5B) to retain the timing information of the behavioral protocols, without showing "hypothetical" behavior.

In its present form, I do not feel that the authors achieved their aims or that the results support their conclusions. Addressing the points above is more likely to produce such results.

We hope that the extensive revision of our paper would be able to convince the reviewer that our paper now reaches its aim. That is, we show, using an array of approaches like *invivo* Ca imaging, *in-vivo* optogenetic manipulations, and *ex-vivo* circuit mapping, that the ventral tail striatum (vTS) shapes learned defensive behaviors during fear learning, by different, but synergistic roles of direct, and indirect pathway neurons in this striatal area.

Although even then, it appears that the contribution of these two cell types to tone-shock learning appears limited.

We agree with the reviewer that the contribution of the ventral tail striatum to fear learning is more of a "modulatory" role, although we clearly describe that the two neuron types have roles during different phases of learned fear behavior. We have made changes in the text to indicate that the role of the vTS in fear learning is rather "modulatory" (see abstract, p. 2, l. 24; conclusions, p. 23, l. 553).

If these neurons are contributing to aversive behavior, perhaps their contribution would be better captured by procedures in which movement is a more central element to behavior.

Thank you for this interesting proposal. We think that developing alternative behavioral procedures for studying the role of the vTS is, unfortunately, beyond the scope of the present paper.

[Editors' note: further revisions were suggested prior to acceptance, as described below.]

The manuscript has been greatly improved during the revision, but there are some issues that remain to be addressed prior to acceptance, as outlined below:Reviewer #2 (Recommendations for the authors):The authors have greatly altered and improved the manuscript to address the reviewer concerns. In particular, they have implemented a standardized Ca analysis pipeline, altered the description of the conclusions in many places, improved the presentation of the data and added some necessary controls.I wish that they had done a full characterization of the RV controls, instead of showing some selected images, but that is ok.

For the first revision, we have added additional control experiments for three conditions (see Figure 7 —figure supplement 1, [panels A, B]: expression of the helper constructs alone in *Drd1a^Cre^* mice, [panels C, D]: expression of the same helper constructs in C57Bl/J mice; [panels E, F]: injection of the rabies virus into the vTS of a *Drd1a^Cre^* mouse in the absence of previous helper virus expression). These three experiments have shown the expected results; so we are not quite sure of the statement by the reviewer "*full characterization of the RV controls*". Nevertheless, the reviewer finally seemed satisfied with the added data to Figure 7 —figure supplement 1 (for Revision 1).

I am confused about the sentence (lines 199-201)"These experiments show that a substantial sub-population of Adora+ neurons in the vTS codes for movement onset, but this representation was unchanged by fear learning (Figure 3M – P)."The changes in event frequency in Figure 3R from habituation to training for freeze state are quite large yet the conclusion is that training does not influence the representation of movement in Adora2a+ neurons? Perhaps I am misinterpreting the statement.

We appreciate that the reviewer seems to observe some differences in the ca^2+^ event frequency data in *Adora2a^Cre^* mice induced by fear learning; however, statistical analysis shows that possible differences are not significant.

The reviewer refers to the ca^2+^ event frequency data shown in Figure 3R. Here, ca^2+^ event frequencies were plotted as average ± S.E.M. values (as indicated in the legend). In the corresponding Figure 3 —figure supplement 2, the same data are plotted as individual data points, and the median, and the interquartile ranges are superimposed on the data (as also indicated in the corresponding Figure legend). Because some of the data is highly nonnormally distributed, it is important to look at the more "raw" data display in Figure 3 —figure supplement 2 to fully understand whether specific data sets are different, or not. In addition, one needs to consult the results from statistical testing, reported in "Source Data Figure3.xlsx".

Specifically, the reviewer states "*that changes in event frequency in Figure 3R from habituation to training for freeze state are quite large*". For example, one might think that the average ca^2+^ event frequency for "Frz_noCS" on the Habituation day (leftmost blue symbol in Figure 3R) is higher than the corresponding value on the training day (third blue symbol in Figure 3R). However, when inspecting the corresponding data and their distributions in Figure 3 —figure supplement 3 ("Frz_noCS" on Habituation day versus "Frz_noCS" on the Training day), it is seen that the medians of these two datasets are almost the same (the average values are different, because the "Frz_noCS" data on the Habituation day shows a stronger skew to high values). Correspondingly, the statistical comparison between the "Frz_noCS" data of the Habituation day, versus the "Frz_noCS" data on the Training day, reports p > 0.99 (Dunn's multiple comparison test; this value can be seen in "Source Data Figure3.xlsx), sheet "Figure 3R cond_activity_Stats", line 33 ("Frz_noCS_Hab vs. Frz_noCS_Train").

Similarly, for other inter-day comparisons within the same movement state, comparisons are non-significant for the ca^2+^ event frequency in the *Adora2a^Cre^* mice. Indeed, the display of the ca^2+^ event frequency data in Figure 3 —figure supplement 3 allows the conclusion that the main condition which changes the ca^2+^ event frequency is "movement" as compared to "freezing" (compare "red" data points with "blue" data points over all three days – all these comparisons are statistically significant as indicated by the "star" symbols above the comparisons).

Thus, we feel that our conclusion statement is justified.

Nevertheless, we have added a side sentence after the above conclusion sentence; the entire sentence now reads (l. 199 – 202):

"These experiments show that a substantial sub-population of Adora+ neurons in the vTS codes for movement onset, but this representation was unchanged by fear learning (Figure 3M – P), except for a decrease in the number of neurons showing a movement – ON response in the absence of a CS (Figure 3P, open symbols)."

The newly added sentence refers to our observation that the number of Adora+ neurons with a movement – ON responses outside the CS decreases with fear learning (data in Figure 3P, open symbols; p = 0.0229; Chi-square test, as reported on l. 196).